# Multiple abiotic stimuli are integrated in the regulation of rice gene expression under field conditions

Anne Plessis[1,2†], Christoph Hafemeister[1,2], Olivia Wilkins[1,2], Zennia Jean Gonzaga[3], Rachel Sarah Meyer[1,2], Inês Pires[1,2], Christian Müller[4], Endang M Septiningsih[3‡], Richard Bonneau[1,2,4*], Michael Purugganan[1,2*]

[1]Department of Biology, New York University, New York, United States; [2]Center for Genomics and Systems Biology, New York University, New York, United States; [3]International Rice Research Institute, Metro Manila, Philippines; [4]Simons Center for Data Analysis, Simons Foundation, New York, United States

*For correspondence: rb133@
nyu.edu (RB); mp132@nyu.edu
(MP)

Present address: †School of
Biological Sciences, Plymouth
University, Plymouth, United
Kingdom; ‡Department of Soil
and Crop Sciences, Texas A&M
University, College Station,
United States

Competing interests: The
authors declare that no
competing interests exist.

Reviewing editor: Daniel J
Kliebenstein, University of
California, Davis, United States

**Abstract** Plants rely on transcriptional dynamics to respond to multiple climatic fluctuations and contexts in nature. We analyzed the genome-wide gene expression patterns of rice (*Oryza sativa*) growing in rainfed and irrigated fields during two distinct tropical seasons and determined simple linear models that relate transcriptomic variation to climatic fluctuations. These models combine multiple environmental parameters to account for patterns of expression in the field of co-expressed gene clusters. We examined the similarities of our environmental models between tropical and temperate field conditions, using previously published data. We found that field type and macroclimate had broad impacts on transcriptional responses to environmental fluctuations, especially for genes involved in photosynthesis and development. Nevertheless, variation in solar radiation and temperature at the timescale of hours had reproducible effects across environmental contexts. These results provide a basis for broad-based predictive modeling of plant gene expression in the field.

## Introduction

Plants have evolved responses to complex environmental fluctuations that take place at time scales that vary from seconds to years and shape plant developmental and physiological responses. Variations in environmental signals, including temperature, water levels, solar radiation, biotic interactions and resource availability, are often unpredictable and need to be integrated and transduced to changes in gene expression, which may then be associated with physiological and/or morphological adaptations (*Ahuja et al., 2010*; *Weston et al., 2008*). Predicting the adaptive responses occurring in natural environments is a key challenge in plant biology. This is an undertaking that will not be achieved without understanding how genes and functional genetic networks are regulated in response to fluctuating stimuli out in nature (*Richards et al., 2009*).

Studies of plant transcriptional responses to environmental perturbations are nearly exclusively undertaken in controlled, static laboratory conditions that are divergent from what is seen in the natural world. While these laboratory experiments have enriched our knowledge of the molecular pathways involved in abiotic stimulus responses, it is clear that organismal phenotypes and the genetic architecture of various traits differ between controlled laboratory and field conditions (*Malmberg et al., 2005*; *Mishra et al., 2012*; *Weinig et al., 2002*). This so-called "laboratory/field (lab-field) gap" is often referred to when trying to explain why the improvement of crops for resistance to abiotic stresses, for example, has not met the expectations arising from advances in

**eLife digest** Plants need to be able to sense and respond to changes in temperature, light levels and other aspects of their environment. One way in which plants can rapidly respond to these changes is to modify how genes involved in growth and other processes are expressed. Therefore, understanding how this happens may help us to improve the ability of crops to grow when exposed to drought or other extreme environmental conditions.

Most previous studies into the effect of the environment on plant gene expression have been carried out under controlled conditions in a laboratory. These findings cannot reflect the full range of gene expression patterns that occur in the natural environment, where multiple factors (e.g. sunlight, water, nutrients) may vary at the same time. Therefore, it is important to also analyze the effect of fluctuations in multiple environmental factors in more complex field experiments.

Plessis et al. developed mathematical models to analyze the gene expression patterns of rice plants grown in the tropical environment of the Philippines using two different farming practices. One field of rice was flooded and constantly supplied with fresh water (referred to as the irrigated field), while the other field was dry and only received water from rainfall (the rainfed field). The experiments show that temperature and levels of sunlight (including UV radiation) have a strong impact on gene expression in the rice plants. Short-term variations in temperature and sunlight levels also have the most consistent effect across the different fields and seasons tested. However, for many genes, the plants grown in the irrigated field responded to the changes in environmental conditions in a different way to the plants grown in the rainfed field.

Further analysis identified groups of genes whose expression combined responses to several environmental factors at the same time. For example, certain genes that responded to increases in sunlight in the absence of drought responded to both sunlight levels and the shortage of water when a drought occurred. The next step is to test more types of environments and climates to be able to predict gene expression responses under future climatic conditions.

genomics technologies (*Cabello et al., 2014*). Given that environmental stresses are one of the main constraints on crop performance, understanding the mechanisms that plants rely on to cope with challenging conditions in the field will be an important asset for crop improvement.

In addition to the dynamic nature of field conditions, the co-occurrence of multiple dynamic signals is another major cause of discrepancies in plant responses/phenotypes between laboratory and field conditions, as laboratory studies generally investigate the short-term effects of single environmental perturbations. The study of plant response to multiple concurrent stimuli, however, can provide insights into both ecological adaptation and, in the context of crop species, crop performance. Concurrent stresses have a major impact: the combination of heat and drought stresses, for example, has been found to be more detrimental to crop yield than the addition of either stress alone (*Mittler, 2006*). The interplay of multiple dynamic factors also affect plants at the physiological level, particularly in the case of photosynthesis: mesophyll conductance depends on both $CO_2$ concentration and irradiance (*Kaiser, et al., 2015*), while drought can undermine the otherwise positive effect of high temperature on photosynthetic efficiency (*Pfannschmidt and Yang, 2012*). Some of the effects of combinations of multiple environmental perturbations on gene expression cannot be predicted from the individual treatments, as has been shown in tobacco (*Rizhsky et al., 2002a*), *Arabidopsis thaliana* (*Rasmussen et al., 2013*) and sorghum (*Johnson et al., 2014*). Because these studies were conducted under controlled conditions, little is known about how the levels and patterns of gene expression are modulated by multiple dynamic abiotic factors in the field (*Izawa, 2015*).

Another feature of plant responses to environmental signals in nature is that they depend on a variety of ecological contexts (e.g. seasonality, macroclimate). Indeed, a reason invoked by plant molecular biologists to avoid experiments in natural environments despite the lab–field gap is the perceived low reproducibility of results that would be generated under unpredictable fluctuating conditions (*Izawa, 2015*) and the difficulty to detect transcriptional signals of interest (*Travers et al., 2007*). These concerns arise mainly from the known sensitivity of gene expression to the overall environmental background, which in nature cannot be controlled. Thus, the extent to which the climatic or ecological context affects molecular environmental responses of interest in

plants remains to be determined. The transcriptional effect of the environmental context is all the more critical considering that, for crop species, different types of agricultural settings or climates will impact dynamic gene expression responses that can translate into key phenotypes.

Understanding both the effect of the environmental context on transcriptomic responses and the integration of multiple stimuli is essential to the development of predictive models for gene expression that can be generalized to wide ranges of agronomical settings and anticipated climatic conditions. A few studies have investigated the dynamic relationships between gene expression and fluctuating environmental conditions in the field in *A. thaliana* (*Richards et al., 2012*), *Oryza sativa* (*Nagano et al., 2012*) and *Andropogon gerardii* (*Travers et al., 2010*) but they provide limited insight on the integration of multiple abiotic stimuli and the effect of environmental contexts on transcriptional responses.

The study we report here is the first to focus on identifying the concurrent effects of multiple environmental factors on gene expression under natural climatic fluctuations in a crop species. Moreover, we examine the macro-environmental context of genome-wide gene expression by measuring transcriptome variation in contrasting seasons and field types. We analyzed the global gene expression patterns in *O. sativa* over a period of 1 month in two fields typical of the main modes of rice cultivation. We conducted these experimentsover two seasons, dry and wet, in three rice landraces. We used model selection methods to relate the main variations in global gene expression to variation in several environmental/developmental parameters with simple linear equations. The genetic background of the plants had limited effect on the environmental response of the global transcriptome. We show that additive effects of several environmental factors drive the field expression of multiple co-expressed gene clusters. We also found that the field context reshaped a large part of transcriptional patterns, while the effect of season was more limited. This sensitivity to the environmental context was particularly important for groups of co-expressed genes involved in photosynthesis and development. Finally, we used previously-published field expression data from *Nagano et al. (2012)* to show that only a small part of the relationships between weather fluctuations and gene expression identified under a tropical climate could be detected in temperate conditions.

## Results

### Isolating environmental effects on gene expression in the field

Our experiment was designed to specifically assess the effect on rice global gene expression of climatic fluctuations, different types of field environments and the genetic background. We conducted two phases of field cultivation - one during the dry season (January–February) and one during the wet season (July–August) at the experimental rice station of the International Rice Research Institute (IRRI), Los Baños, Laguna, in the island of Luzon in the Philippines in 2013 (*Figure 1A*). The plants were cultivated in two adjacent fields typical of two different systems of rice cultivation. One field was cultivated following irrigated lowland practices: it was flooded with shallow water under constant irrigation and seedlings were transplanted to the field after being raised for 3 weeks in seedbeds (referred to as irrigated field). The second field was managed according to upland cultivation practices: it was rainfed, not irrigated and had been directly seeded (referred to as rainfed field). Each field was divided into two subfields constituting the biological replicates. We grew three different landraces of rice: (i) Azucena, an upland adapted landrace; (ii) Pandan Wangi, traditionally used for lowland cultivation only in the dry season; and (iii) Palawan, another upland adapted landrace that is used only during wet season cultivation.

To avoid the major shift in gene expression patterns induced by the transition to the flowering stage (*Sato et al., 2011*), which would confound our detection of environmental effects, we sampled rice leaf tissue during 1 month of vegetative growth (15 sampling timepoints, 2 days apart). Sampling was carried out 4 hr after sunrise to minimize circadian-driven transcriptional variation. Each sample included six young leaves, each from a different plant, to minimize variation in individual plant microenvironment.

We measured global gene expression using RNA sequencing (*Figure 1B*). We excluded from our analysis genes for which we detected sequencing reads for less than 20 samples out of the 60

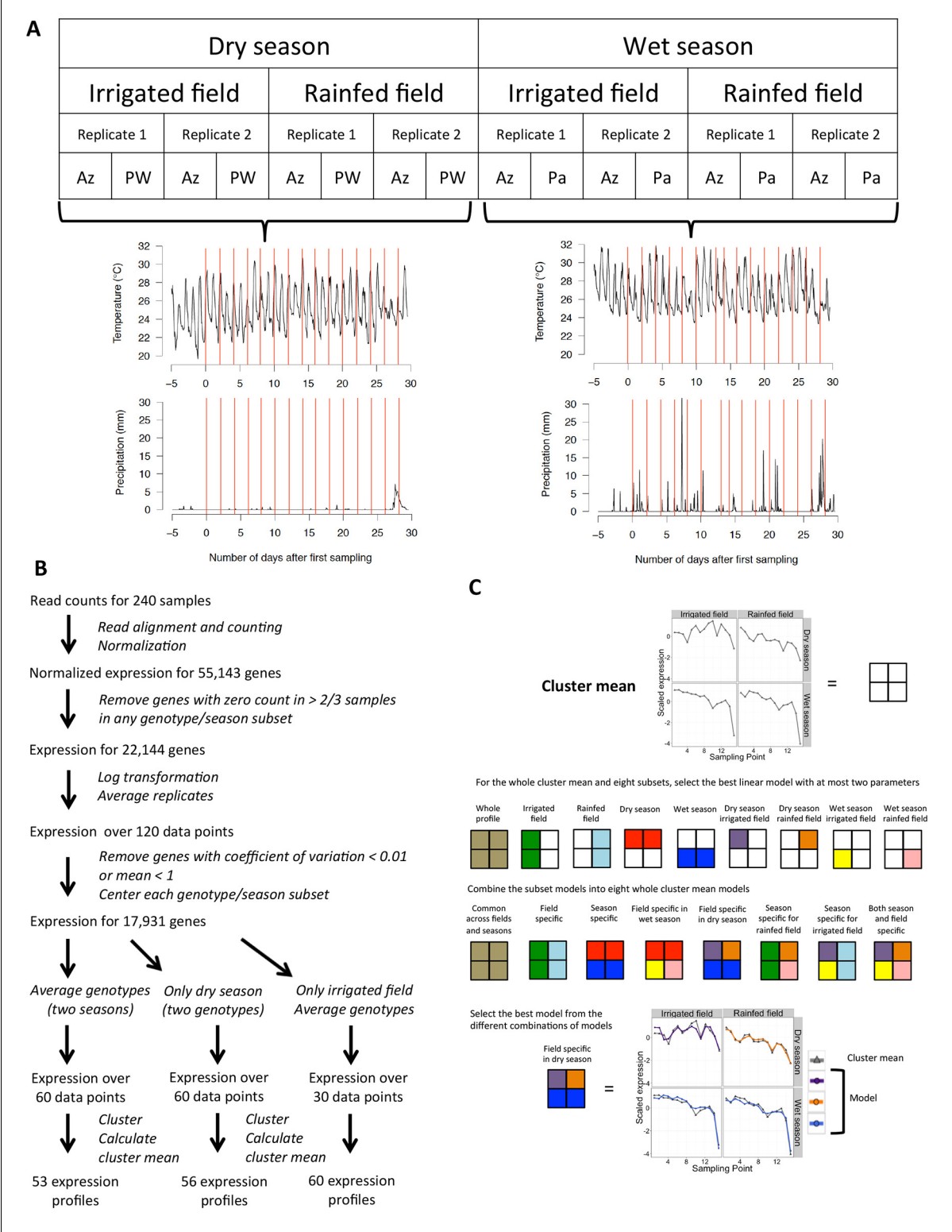

**Figure 1.** Modeling environmental and developmental effects on rice gene expression in the field. (**A**) Experimental design: fifteen sampling timepoints for each of the 16 season/field/replicate/genotype series amounts to 240 samples, representative of 30 different sets of climatic conditions, represented here by the 30 red lines on the graphs for temperature and precipitation (accumulated mm per 5 min) along the sampling period of each growing season. (**B**) Processing raw RNA sequencing data into main transcriptomic variation for different subsets of our data. (**C**) Modeling potential differences between fields and seasons for climatic/developmental response within a cluster mean. An example for cluster 15 of the two-season analysis, where the

*Figure 1. continued on next page*

Figure 1. Continued

model selected is common to both fields in the wet season but field specific in the dry season. Az, Azucena landrace;  Pa, Palawan landrace; PW, Pandan Wangi landrace.

samples in each genotype per season subset of the data. The expression data for the 22,144 remaining genes was log-transformed and the biological replicates were averaged.

## Modeling the effect of climatic factors on transcriptomic variation in different field environments

Our goal is to relate gene expression variation over time to variation in climatic conditions and plant developmental stage, and assess how these relationships are affected by season, field type and genetic background. We focused on trends in gene expression variation common to a high number of genes: after removing 1251 genes with a low coefficient of variation and 2962 genes with a low mean expression, we grouped the remaining genes into co-expressed gene clusters (*Figure 1B*). The number of clusters chosen was the highest that satisfied the constraint that no more than 5% of all the genes in the analysis belonged to "non-representative" small clusters, defined as containing less than 1% of all the genes in the analysis. We used the mean expression profile of all genes in each cluster as a representation of the variation in expression within that cluster.

We used a model selection approach to explain gene expression patterns by environmental and developmental variation. This approach relies on selecting a linear combination of environmental/developmental (ED) input parameters that both minimizes model mean squared error (MSE), quantifying the difference between the model and the expression data, and limits model complexity (i.e., avoiding over-fitting). A preliminary analysis showed that allowing for more than three parameters per equation over-fit the model more often than improving it, so we limited the number of parameters per linear equation to three. A typical ED equation had the following form: cluster mean =

**Table 1.** Climatic parameters for the environmental/developmental models and their abbreviations. Parameters calculated from the weather data are mostly ordinary averages (linear: L) for a large range of time windows before sampling but also include exponentially transformed averages (non-linear: NL) modeling stronger effects at either low (NL-) or high (NL+) values. For the most dynamic climatic parameters, we calculated differences (D) between the measurement at the sampling timepoint and the measurement short amounts of time before sampling. To estimate fluctuations, we calculated averages of the residual term (R) from the seasonal decomposition of daily variation into cyclic and trend components. The abbreviation for each climatic factor and parameter type and time-window is given in parentheses.

| | Average/change for the last | Temperature (tp) | Relative humidity (hu) | Solar radiation (so) | Wind speed (wd) | Atmospheric pressure (ps) | Rainfall (ra) |
|---|---|---|---|---|---|---|---|
| Sampling time | 15 min (15 min) | L | L | L NL− NL+ | L | L | L |
| Short-term averages | 1 hr (1 hr) | L | L | L NL− NL+ | L | L | L |
| | 4 hr (4 hr) | L | L | L NL− NL+ | L | L | L |
| | 24 hr (24 hr) | L | L | L | L | L | L |
| Long-term averages | 3 d (3 d) | L | L | L | L | L | L |
| | 6 d (6 d) | L | L | L | L | L | L |
| | 10 d (10 d) | L | L | L | L | L | L |
| | 15 d (15 d) | L | L | L | L | L | L |
| Recent change | 20 min (δ20 min) | D | D | D | D | | |
| | 1 hr (δ1 hr) | D | D | D | D | | |
| | 2 hr (δ2 hr) | D | D | D | D | | |
| Fluctuations | 1 hr (ε1 hr) | R | | | | | |
| | 4 hr (ε4 hr) | R | | | | | |
| | 24 hr (ε24 hr) | R | | | | | |

αED1 + βED2 + γED3, where ED1, ED2 and ED3 are ED parameters, and α, β and γ are linear regression coefficients. The ED parameters used in these models were measurements of current conditions at the time of sampling, recent changes in temperature, humidity, wind speed and solar radiation, temperature fluctuations, and short-term and long-term averages for all climatic conditions (*Table 1*). We included parameters for non-linear effect of short-term solar radiation on gene expression, because this type of effect has been observed on photosynthesis rate (*Li et al., 2009*). There were two parameters for field soil moisture, at 30 and 15 cm below ground (measured with tensiometers in the rainfed field and estimated to be constant at soil saturation value in the irrigated field), and a binary parameter for the field (irrigated or rainfed). A parameter was designed to represent developmental stage, using fixed values for the transplanting stage, end of tillering production and the heading time. Our set of ED parameters included those that correlated with each other, so we averaged nearly similar parameters (r > 0.98) and added to our model selection approach the constraint that two parameters with a Pearson correlation coefficient over 0.85 could not be selected in the same equation.

We designed our approach to take potential differences in climatic response between fields, genotypes and seasons into account. To assess whether disparities in transcriptional patterns could be explained using distinct ED equations, we considered different ED models for each cluster. The simplest model is a single equation for the whole cluster mean. A more complex model would combine two different equations, for example, in the case of a field-specific model: cluster mean = $\alpha_i$ED1 + $\beta_i$ED2 + $\gamma_i$ED3 in the irrigated field and cluster mean = $\alpha_r$ED4 + $\beta_r$ED5 + $\gamma_r$ED6 in the rainfed field.

If we were to take into account all possible differences between fields, genotypes and seasons, we would get models with as many as eight equations, which would be difficult to interpret. We therefore limited the maximum number of equations to four by applying the method to one season at a time, testing different equations between fields and genotypes; or considering only one genotype and examining the possibility of different equations between fields and seasons (this latter case is represented in *Figure 1C*). We chose between the models comprising one to four equations using the Bayesian Information Criterion (BIC), a statistical tool to limit over-fitting that includes a complexity penalty, calculated on the global model for all subsets.

## Modeled environmental responses are shared by both genotypes in the dry season: a reason to average genotypes

We first used our model selection approach on the dry season data alone, as this was the season with the greatest phenotypic differences between the rainfed and irrigated fields. We wanted to determine the extent of genotype and field differences in gene expression variation during that season and how well the ED models can explain these differences. For a given cluster, we quantified the differences in variation of gene expression between the two genotypes by calculating the correlation between the two genotype-specific subsets of the cluster mean. The same method was used to evaluate differences in expression patterns between fields.

Only 19 out of the 56 clusters (4663 genes) had a correlation coefficient between genotypes below 0.8 (*Figure 2A*). Clusters with a low genotype correlation had a high model MSE, which showed that the ED models did not adequately explain these genotype differences. We found much more extensive differences in gene expression patterns between field environments (*Figure 2B*), with 48 clusters (15,103 genes) that had a correlation coefficient below 0.8. While all clusters with high correlation between field environments (r > 0.8) had ED models that fitted the cluster mean well (model MSE < 0.12), low MSE models were also selected for several clusters with strong dissimilarities between field environments. In some cases, ED models could thus explain strong field differences. We did not investigate the genotype effect further and instead used the genotypes as biological replicates that were averaged for the analysis of both seasons, incorporating all the expression data concatenated into 60 data points (*Figure 1A*).

## Simple models can explain differences in transcriptomic patterns between fields across two seasons

In this two-season analysis, in addition to identifying which ED parameters the expression of each gene cluster can be related to, we are assessing whether the field environment and the season affect the identified transcriptional response. As an example, the simple season-specific ED model selected

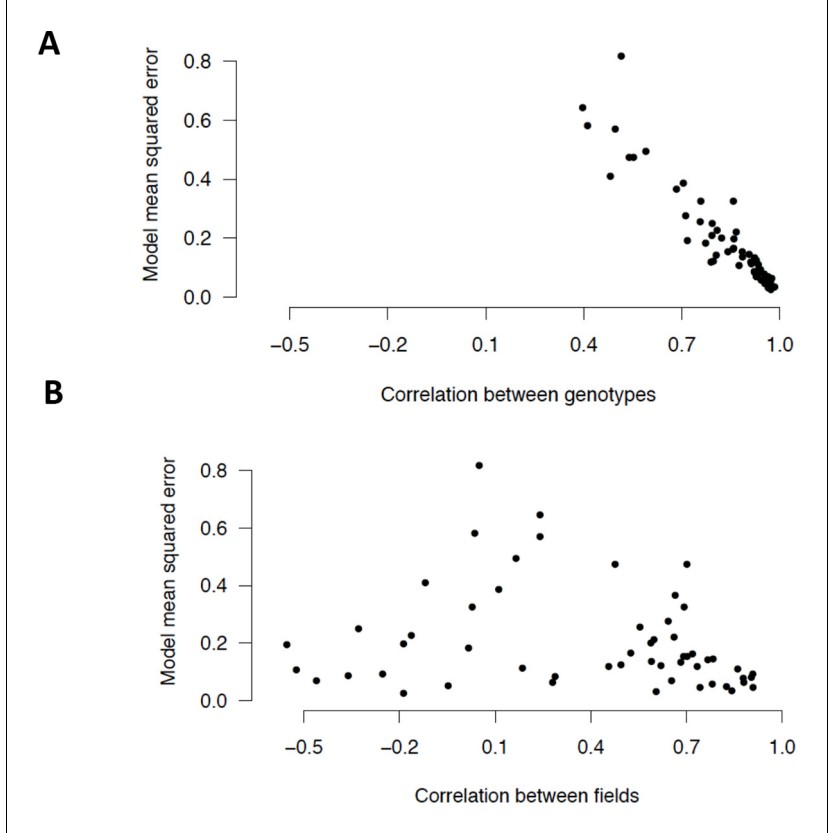

**Figure 2.** In the dry season analysis, expression patterns of the two genotypes are highly correlated for most clusters while for the few other clusters, differences between the genotypes are poorly explained by the models (high model MSEs for clusters with low correlation). (A) Model MSE vs. correlation (Pearson coefficient) between the genotypes for the 56 dry season cluster means. (B) Model MSE vs. correlation between the fields for the 56 dry season cluster means. MSE, mean squared error

for cluster 9 is presented in *Figure 3A* (*Supplementary file 1A–C*). The model consists of a single linear equation for both fields and both seasons. It combines a negative term for soil moisture at 15 cm depth, a positive term for 1-hr average of solar radiation (exponentially transformed) and a positive term for the change in temperature during the last 2 hr. In this model, the soil moisture input parameter (*Figure 3B*, *Supplementary file 1B*) allows for modeling gene expression differences between field environments while keeping a common model for both field environments, in particular the higher expression in the rainfed field environment in the dry season. The selection of a common model for the dry and wet seasons shows that the gene expression response to climatic factors of this cluster was largely independent of the season.

We used genotype correlation within a cluster as a measure of replicability of ED effects: the higher the genotype correlation, the more gene expression response was driven by factors common to both genotypes (i.e., climatic and developmental factors). While the median genotype correlation of all genes in the analysis was 0.55, the median of all gene cluster means was 0.90, showing that averaging expression profiles over many genes remarkably reduces sources of non-replicability. We focused on 27 gene expression clusters with a genotype correlation greater than 0.9 (*Figure 4*, *Supplementary file 1C*). They encompass 11,371 (63%) of the 17,931 genes in the analysis, as they include most of the largest gene clusters. These clusters are also the ones with the best ED models (i.e., with a small error and low complexity, such as the one depicted in *Figure 3*, *Supplementary file 1C*). For these clusters, we observed extensive field effect on the climatic/developmental response, with only 6 out the 27 clusters (2308 genes) showing high correlation (r > 0.8) between fields in both seasons. As expected from the much lower precipitation levels during the dry season, which accentuated the difference in water availability between fields (*Figure 1A*), the dry

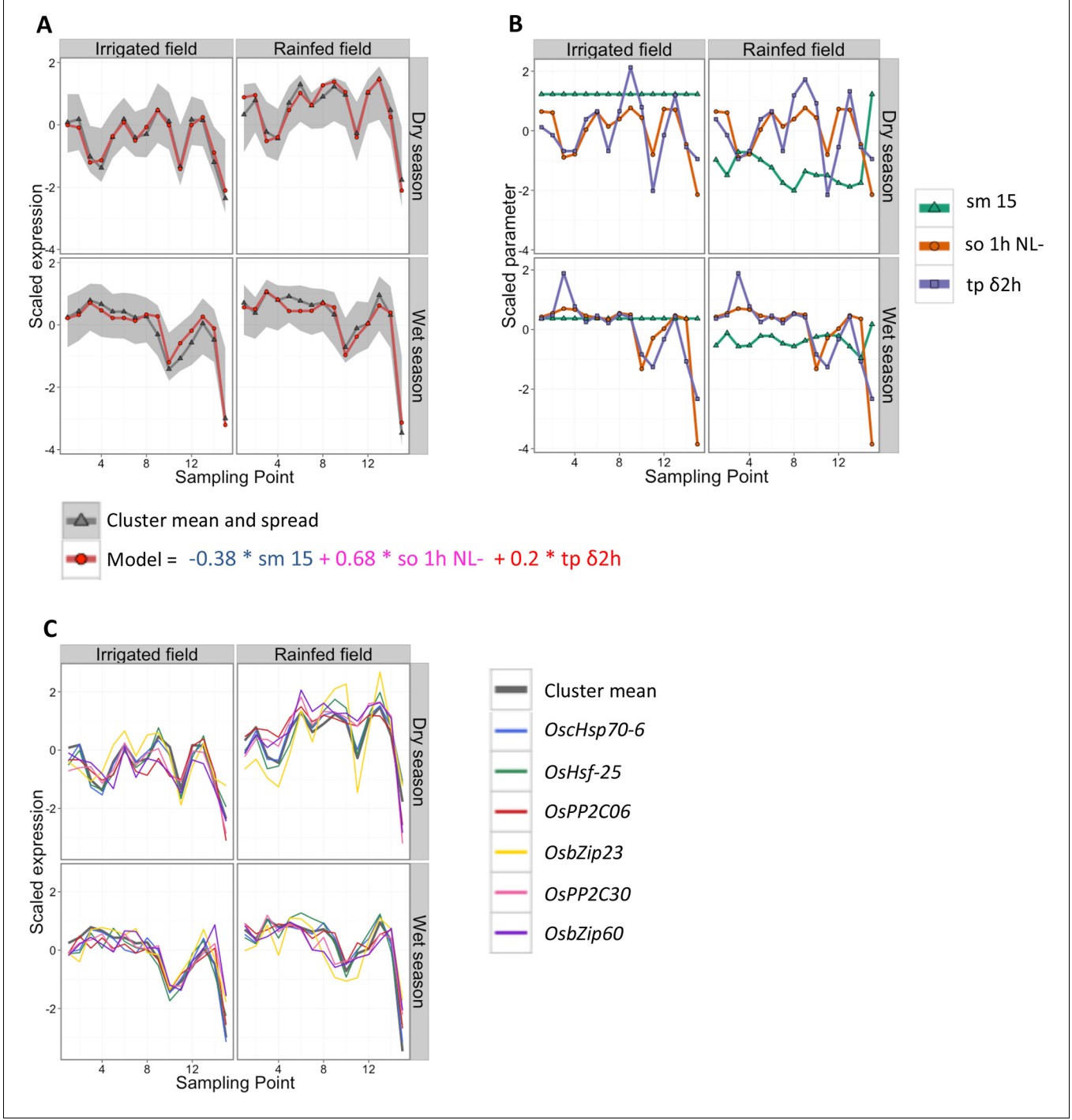

**Figure 3.** Cluster 9 of the two-season analysis: its environmental/developmental model and some of the genes it contains that have a potential function in environmental response. (**A**) Gene expression for the cluster mean (grey) and spread (calculated as 10% and 90% quantile of all genes in the cluster for each data point; grey area), and cluster model (red). (**B**) Scaled climatic parameters in the model equation for each season. sm 15: soil moisture at 15 cm; so 1h NL-: one hour average of solar radiation transformed to increase the effect of low values; tp δ2h: change in temperature from 2 hr ago. (**C**) Gene expression pattern (scaled) of six genes in cluster 9 with a potential function in environmental response.

season climatic conditions generated wider between-field differences in gene expression compared with the wet season (*Figure 4*, *Supplementary file 1C*). The season effect was slightly less prevalent,

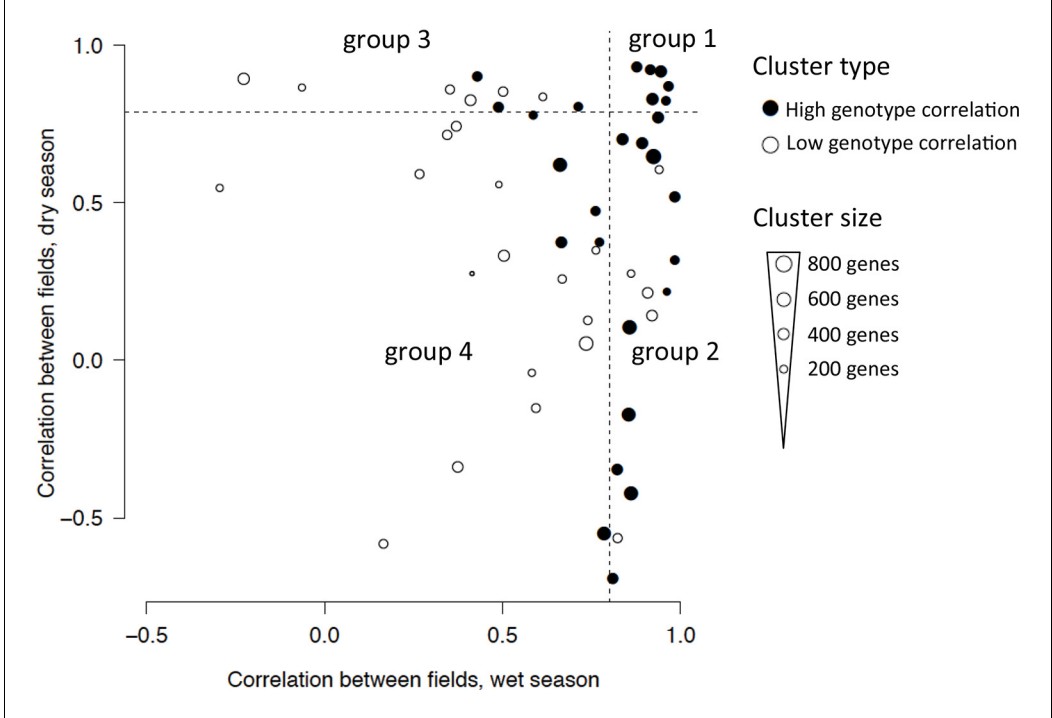

**Figure 4.** Classification of the 53 clusters from the two-season analysis based on field correlation in the dry and wet seasons. Each dot represents a cluster with the size of the dot proportional to the number of genes in the cluster. The 27-gene clusters with igh genotype correlation (r > 0.9, Pearson coefficient) are represented with a filled circle and divided into four groups depending on whether their correlation between fields in each season is below or above 0.8 (dashed lines), referred to as high or low correlation. Group 1 clusters have a high correlation between fields in both seasons; group 2 clusters have a high correlation in the wet season but not in the dry season; group 3 clusters have a high field correlation in the dry season but not in the wet season; group 4 clusters have a low field correlation in both seasons.

as 11 out the 27 clusters (4967 genes) were modeled with the same equation for both seasons (*Supplementary file 1C*), indicative of season-independent responses.

## Field environment strongly impacts the transcriptional regulation of photosynthesis and development

We analyzed gene clusters according to their field response to understand how distinct modes of cultivation affect gene expression under the same climate. The expression of a gene cluster can be affected by the field environment in two ways: (i) distinct responses to climatic/developmental factors and/or (ii) a shift in expression level, representing different ways in which the effect of the field environment can be integrated with the climatic response and developmental program. Enrichment in specific functions or pathways within the different types of gene clusters can be indicative of the role of certain processes in the adaptation to distinct field environments.

We divided the 27-gene clusters into four groups based on whether they showed different expression responses to climatic/developmental factors in each of the two fields (correlation between the expression patterns of the two fields below 0.8) in one or two seasons (*Figure 4*). We also calculated the difference in mean expression between the fields for each season. To investigate the molecular processes that were most affected by climatic variation and field environments, we conducted a gene ontology (GO) term enrichment analysis for each cluster (*Supplementary file 1E*).

Correlations between ED parameters make it difficult to ascertain the causal factor of gene expression change from the parameter selected in a model. This is why, for the interpretation of ED models, we grouped those of the parameters that showed high correlations to each other (*Figure 5A*). The parameters (sometimes included in a group of highly correlated parameters) selected for the 27-gene clusters most representative of ED responses are shown in *Figure 5B*.

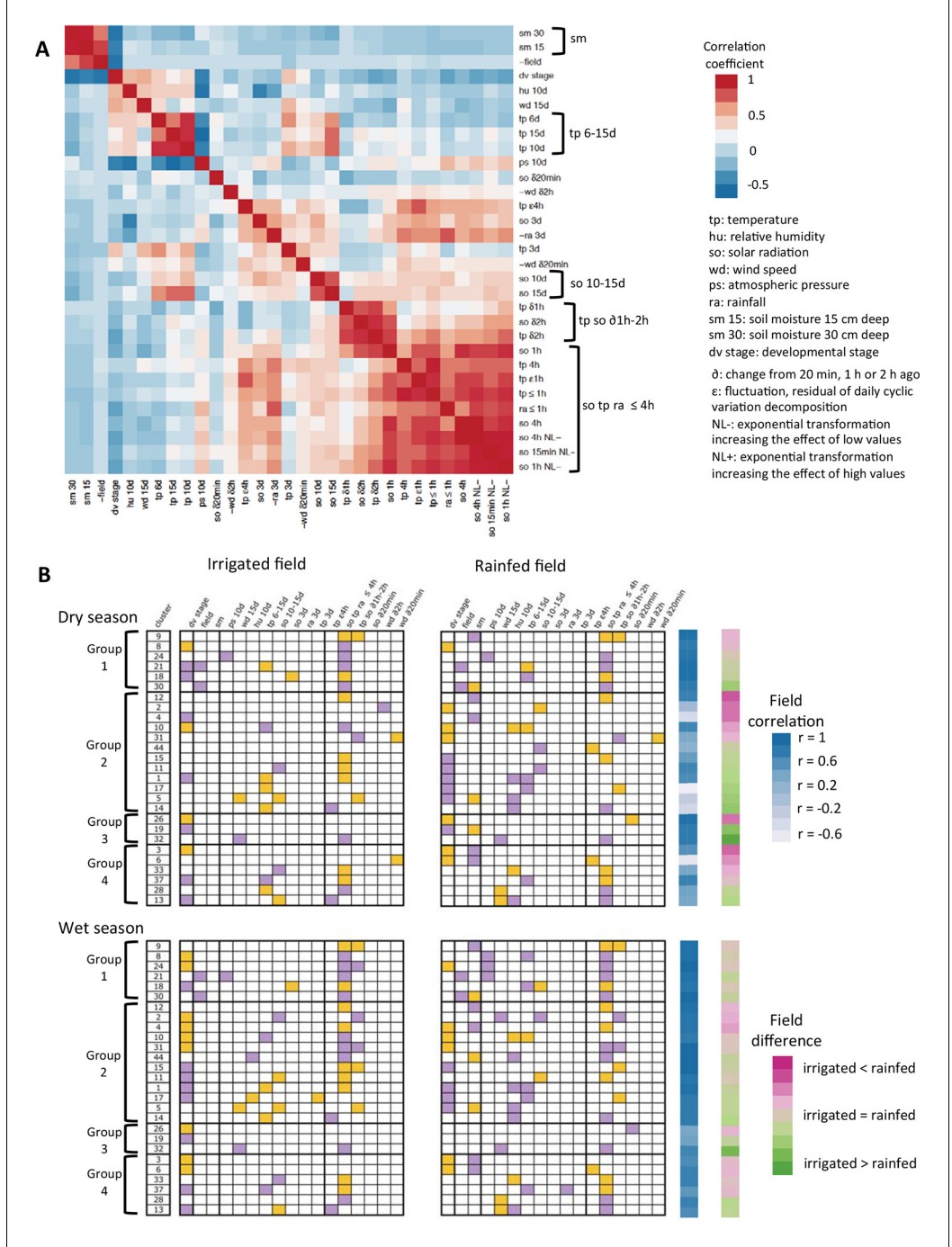

**Figure 5.** Summary of the ED models selected for the 27-gene clusters with high genotype correlation in the two-season analysis. (**A**) Justification for the grouping of model parameters: heat-map of the correlation between the ED parameters selected at least once in the models. For some parameters that have strong negative correlations with other parameters, we used negative values to better see groups of correlated parameters independently of the sign of the correlation. (**B**) The selection of a parameter into a model is represented by a colored box, orange for a positive term and purple for a negative term. Field correlation is the correlation between irrigated and rainfed field profiles for each season. Field difference is the average expression level in the rainfed field minus the average expression level in the irrigated field. ED, environmental/developmental

## Similar patterns across field environments were driven by short-term climatic fluctuations

The field environment had the least effect on the climatic/developmental expression response of the six clusters in group 1 (*Figure 4*). Nevertheless, some of these clusters showed shifts in expression between the fields in the dry season (*Figure 5B*, *Supplementary file 1C*). They were mostly influenced by short-term climatic conditions and, to a lesser extent, developmental factors (*Figure 5B*, *Supplementary file 1C*), suggesting that short-term temperature/solar radiation had a strong effect on gene expression variation over time that could be independent of the field environment even under limited water availability.

Gene clusters 9 and 18 are both modeled with positive terms for short-term temperature/solar radiation across both fields and both seasons but are differently affected by the field environment. While cluster 18 displays a higher expression in the irrigated field than the rainfed field in the dry season, cluster 9 shows an increase in the mean expression level in the rainfed field compared with the irrigated field in both seasons, but more distinct in the dry season. This field effect is modeled for cluster 9 by the inclusion of a negative term for soil moisture, indicative of an additive effect of water availability on gene expression over the short-term effect of temperature/solar radiation (*Figure 3A and B*). Cluster 9 is highly enriched for genes related to response to heat ($p<10^{-18}$, hypergeometric test), response to high light intensity ($p<10^{-9}$) and, more generally, response to abiotic stimulus ($p<10^{-6}$), which is consistent with an induction by drought, solar radiation and/or temperature increase. The four gene clusters from group 1 with models that have negative terms for short-term temperature/solar radiation are enriched for signaling/regulatory categories of genes – cluster 8 for response to hormone stimulus, cluster 21 for cell communication and signaling, cluster 24 for protein kinase activity and hormone-related signaling pathway and cluster 30 for regulation of transcription, DNA-dependent. This shows that short-term temperature/solar radiation effects on the expression of groups of co-expressed abiotic stress response and signaling/regulatory genes can be impervious to the field environment.

## Drought-affected climatic and developmental response of abiotic response, photosynthesis and developmental genes

The 12-gene clusters in group 2 have in common strong differences in gene expression pattern between the rainfed and irrigated fields that are specific to the dry season. These differences are probably due in great part to the drought period experienced during that season. For these clusters, the field environment during the dry season affected both climatic/developmental responses and mean gene expression level (*Figure 5B*, *Supplementary file 1C*). Gene clusters in which expression was higher in the rainfed field (clusters 2, 4, 10, 12 and 31) showed an overall increase of expression over time in that field environment (*Supplementary file 1A*), consistent with an induction of expression following the decrease in soil moisture. The opposite observation was true for gene clusters showing lower expression levels in the dry season rainfed field (clusters 1, 5, 11, 14, 15, 17 and 44).

This steady increase/decrease in gene expression in the dry season rainfed field was modeled in some cases with a soil moisture term (clusters 4, 5 and 12) consistent with a drought effect. However, parameters for long-term averages of solar radiation, temperature, humidity or even developmental stage were also selected. All of these parameters vary somewhat monotonically over the course of the dry season (*Supplementary file 1C*) but differ from the soil moisture parameter in that they do not include a sharp increase/decrease in their values at the last timepoint of the dry season (where sampling was carried out in heavy rain). These climatic parameters may thus model a pattern of drought response that does not imply short-term recovery.

Among the gene clusters with a lower level of expression in the rainfed field, clusters 1 and 5 are strongly enriched for genes associated with photosynthesis. Both clusters are also enriched in genes related to lipid biosynthetic process and for cluster 1, secondary metabolic and carbohydrate catabolic processes. This result is consistent with the deleterious effect of abiotic stress on photosynthesis (*Chaves et al., 2009*) and shows the consequences of that effect on the expression of genes involved in the metabolic processes dependent on photosynthetic products. In contrast, cluster 12 had a higher level of expression in the rainfed field environment in the dry season. Like cluster 9 in group 1, it is modeled with positive terms for short-term temperature/solar radiation across both fields and both seasons and is enriched for several categories of abiotic stimulus response genes.

Cluster 12 differs from cluster 9 in that the difference in gene expression between fields during the dry season is more acute, with a higher difference in mean expression and a lower field correlation. This shows that there were two distinct main behaviors of abiotic stress response genes in our experiment, both responding to short-term temperature/solar radiation and drought but to different extents. Clusters 4 and 31 also have an expression pattern consistent with induction by drought. They are both enriched for genes in categories related to development and phenology, like cell cycle for both, anatomical structure development and photomorphogenesis for cluster 4 and photoperiodism for cluster 31.

## Field environment affected the expression of clusters enriched for developmental and photosynthesis genes in both seasons

Gene clusters in groups 3 and 4 show differences between the field environments during the wet as well as the dry season, which suggests that their expression is affected by aspects of the field environment not related to water availability. Gene clusters in group 3 had different mean expression level in the two fields in the dry season and a low correlation between the two field environments in the wet season (*Figure 5B*, *Supplementary file 1C*). Group 4 clusters also had strong field differences; they had low correlation in gene expression and differences in mean level of expression between fields in both seasons. Some gene clusters in groups 3 and 4 show strong functional enrichment. Cluster 6 is enriched for regulation of shoot development and developmental process, while cluster 13 is enriched for photosynthesis genes. We found that several clusters in groups 3 and 4 were modeled with terms for long-term wind speed, in both fields and seasons in the case of cluster 32. Interestingly, this cluster is enriched for genes related to thigmotropism, the directional response to mechanical stimuli.

## The transcriptional regulation of characterized environmental response genes integrates multiple abiotic stimuli

The functional characterization of genes involved in the response to abiotic stresses is mostly conducted with single environmental perturbations under controlled conditions. Few genes have been functionally characterized for a role in the response to combined environmental stresses. We have found multiple clusters with patterns of expression showing additive effects of several environmental signals. Because changes in the expression of a gene in reaction to perturbations in an environmental factor are usually interpreted as this gene having a function in the physiological response to that factor, our results suggest the genes in these clusters are involved in the response to simultaneous environmental changes. We assessed this further by looking at whether genes shown to have a role in the response to a specific abiotic stimulus also responded transcriptionally to other environmental signals. We compiled lists of genes that had already been demonstrated to have a role in the response to light, water availability or temperature in rice as well as rice homologs of such genes in other species (mainly the model system *A. thaliana)*. We looked at whether the expression patterns of these genes were consistent with their putative function. We focused our attention on the candidate genes that showed a replicable environmental response in our data (genotype correlation above 0.8) and belonged to one of the 22 gene clusters whose main ED model terms are environmental parameters (*Supplementary file 1C*), which we refer to as genes with a strong environmental regulation.

### Light response genes

From several published reviews, we identified 273 genes involved in light response (*Galvão and Fankhauser, 2015*), in particular phytochrome signaling (*Wang and Wang, 2015*), phototropin signaling (*Sullivan et al., 2009*), retrograde signaling (*Häusler et al., 2014*; *Vogel et al., 2014*), UV signaling (*Singh et al., 2014*), brassinosteroid biosynthesis and signaling (*Zhang et al., 2014*) and the regulation of photosynthesis in response to light (*Kaiser et al., 2015*; *Pfannschmidt and Yang, 2012*; *Rochaix et al., 2012*). Out of the 45 genes with a strong environmental regulation (*Figure 6A*, *Supplementary file 1F*), 31 were in nine clusters modeled with positive terms for short-term temperature/solar radiation while 14 were in seven clusters modeled with negative terms for short-term temperature/solar radiation. Most of these genes were also affected by the field environment, especially in the dry season. However, one gene from cluster 15 was highly variable over time

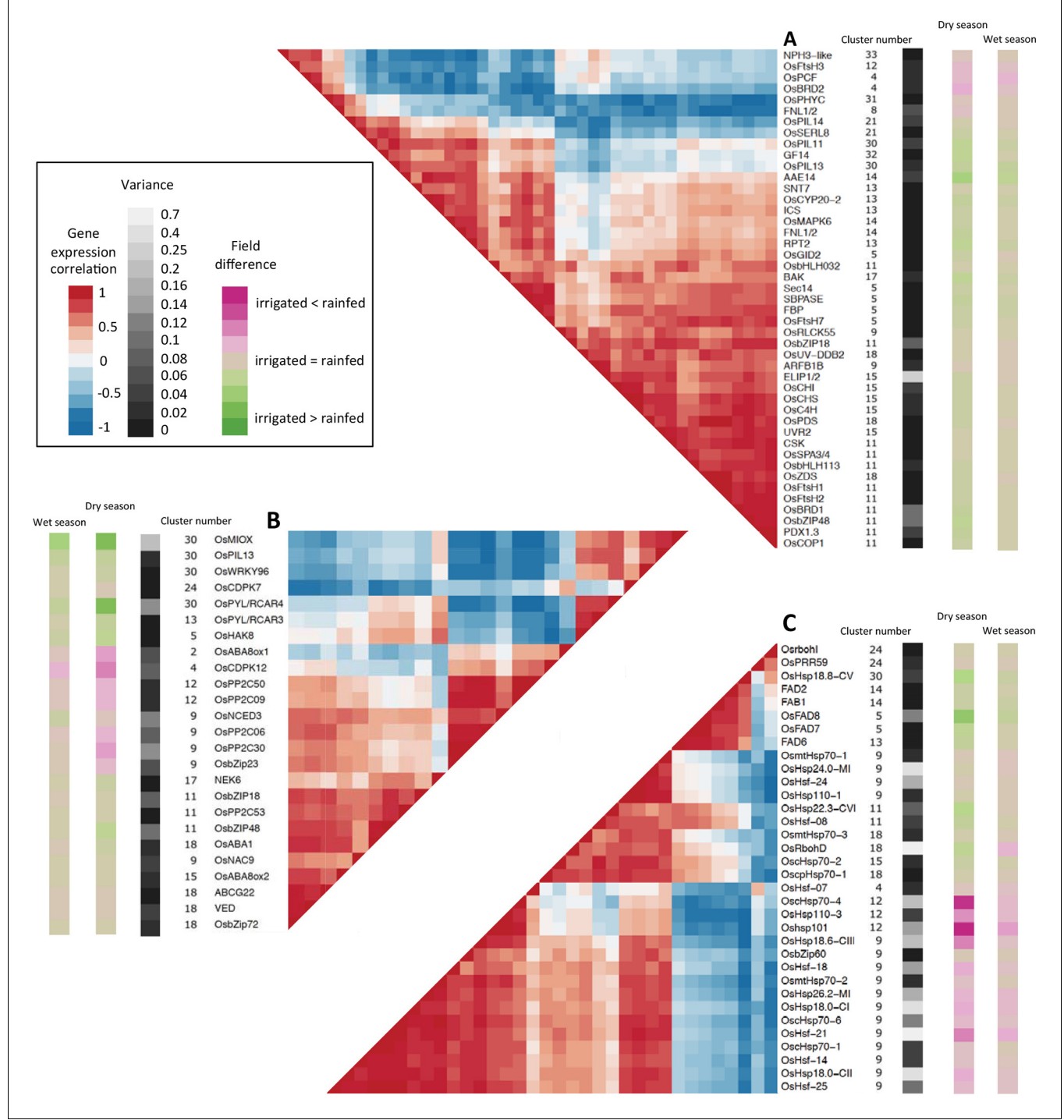

**Figure 6.** Co-expression and variance of genes involved in environmental response. High variance is represented with lighter shades of gray. The genes represented have a genotype correlation above 0.8. Gene names starting with "Os" are rice gene names, others are the names of *A. thaliana* orthologous or homologous genes or gene families. Field difference is average expression level in the rainfed field minus average expression level in the irrigated field. Variance is the mean of the expression variance of each gene across the four season/field environment subsets. (**A**) Genes potentially involved in response to light. (**B**) Genes potentially involved in response to drought. (**C**) Genes potentially involved in response to temperature.

(it belonged to the 95[th] percentile for variance among the 17,931 genes in the 53 clusters) but not between fields. It is an ortholog of *A. thaliana ELIP1* and *2* genes, which are possibly involved in pigment accumulation in response to stress (*Casazza et al., 2005*; *Rossini et al., 2006*).

A group of co-expressed genes included a zeta-carotene desaturase and a phytoene desaturase, both carotenoid biosynthesis enzymes (*Fang et al., 2008*), two *FtsH* proteases and an ortholog of the chloroplast sensor kinase (*CSK*) all of which have a potential role in photoprotection. *OsCHS*, *OsC4H* and orthologs of *A. thaliana UVR2* and *PDX1.3*, four genes that could be involved in UV acclimation (*Singh et al., 2014*), were also co-expressed with the photoprotection genes. These results show that transcriptional regulation is an important component of light response, especially for photoprotection and UV acclimation, and integrates responses to other abiotic stimuli linked to the field environment.

## Drought response genes

We compiled a list of 79 genes potentially involved in drought response in rice, including genes that were used to engineer drought-tolerant plants (*Todaka et al., 2015*). Because abscisic acid (ABA) is an essential component of drought response, we also included 46 genes associated with ABA biosynthesis, catabolism, transport and signaling (*Figure 6B*, *Supplementary file 1F*). Out of the 25 genes from this list with a strong environmental regulation, 10 belonged to clusters in group 2 (*Figure 5B*) with a low field correlation only in the dry season, indicating a drought effect on expression and one gene belonged to a cluster in group 4, and showed more difference in mean expression between fields during the dry season. Fourteen genes belonged to clusters in group 1 with correlated patterns of expression over the two fields, but half of these genes showed a distinct difference in mean expression between irrigated and rainfed field in the dry season. This included genes encoding the protein phosphatases *OsPP2C06* and *OsPP2C30* and the transcriptional regulator *OsbZip23* in cluster 9, the expression profiles of which are represented in *Figure 3C*. All 25 genes belonged to clusters modeled with parameters for short-term temperature/solar radiation averages or changes. In the case of gene clusters 2, 5, 4 and 11, however, no temperature/solar radiation parameter was selected for the model of the rainfed field in the dry season, suggesting that these short-term climatic responses were overridden by a drought response. Seven of the candidate genes displayed weak transcriptional response to the drought period in the dry season, and belonged to clusters modeled with a positive term for short-term temperature/solar radiation. These genes could be responding to abiotic stresses other than drought (e.g. heat, high light) in our conditions.

Our analysis thus indicates that genes involved in drought response may also respond to short-term temperature/solar radiation conditions. For the genes we examined, a transcriptional response to variations in water availability was not always detected, but when it existed, it could be in addition to the short-term climatic response or dominate under drought conditions.

## Temperature response genes

We compiled a list of 151 genes with a potential role in temperature response (*Penfield, 2008*; *Wigge, 2013*) plus 80 known heat shock factors and heat shock proteins. Twenty-eight of the 34 genes with a strong environmental regulation belonged to seven gene clusters modeled with positive terms for short-term temperature/solar radiation. Several heat shock (HS) genes in clusters 9 and 12 had a greater variation over time and between fields than any selected drought-related gene (*Figures 6B and 6C*, *Supplementary file 1F*) while the expression of several HS genes in cluster 18 was not affected by the field environment. Our results show that, in the conditions of our experiment, genes involved in temperature response were mostly induced by short-term heat or light and suggest the frequent transcriptional integration of this response with changes in expression linked to water availability.

## Transferability of climatic effects to an independent dataset of rice gene expression in the field

Here we examine the correspondence of gene expression and climatic variation between our Philippine experiment and a previously published rice field transcriptomic study conducted in different (temperate) climatic conditions (*Nagano et al., 2012*). From the latter experiment conducted in Japan, we selected 52 daytime timepoints that spanned nearly 7 weeks of vegetative growth in irrigated fields for the Japonica cultivar Nipponbare. We centered (subtracted the mean) the data by time of day to eliminate circadian clock effects. We used the same climatic measurements as those

found in our experiment, which were recorded every minute. We refer to this data as the partial Nagano dataset (PND). To determine to what extent our results would hold true under a different climate, we performed a new analysis of our data that excluded measurements from the rainfed field (using the irrigated field half of the two seasons analysis dataset), as the experiment in Japan was exclusively conducted in an irrigated field (*Figure 1B*). We used these results to test whether ED parameters selected in the models for our data could also explain expression variance of the same clusters of genes in the PND.

The models selected for the 60 gene cluster means calculated from our data (*Supplementary file 1G*) could be represented by a single equation for the irrigated field in both seasons (season-independent) or of distinct equations for the irrigated field in each season (season-specific). We reasoned that climatic response patterns that were not consistent between the two seasons of our experiment (season-specific models) were unlikely to be reproduced in the PND. Consequently, we only tested the transferability of models for those of our clusters with a season-independent model, when the model explained more than half of the variance of the cluster mean. For each of these 36 clusters, we calculated the mean expression profile in the PND of the genes in the cluster. We determined whether one or several parameters in the model of the cluster mean over our irrigated field data could model the cluster mean over the PND with the same coefficient signs, by comparing all possible models with combinations of these parameters with the BIC.

Non-null models were selected for 24 of the 36 PND cluster means (*Figure 7*). They explained between 13 and 86% of the expression variance over the 52 data points of the PND. The parameters that could explain expression variation in our experiment and the PND were predominantly developmental stage and parameters for short-term solar radiation and temperature, especially in the models that explained a large part of the expression variance. Only one long-term climatic parameter

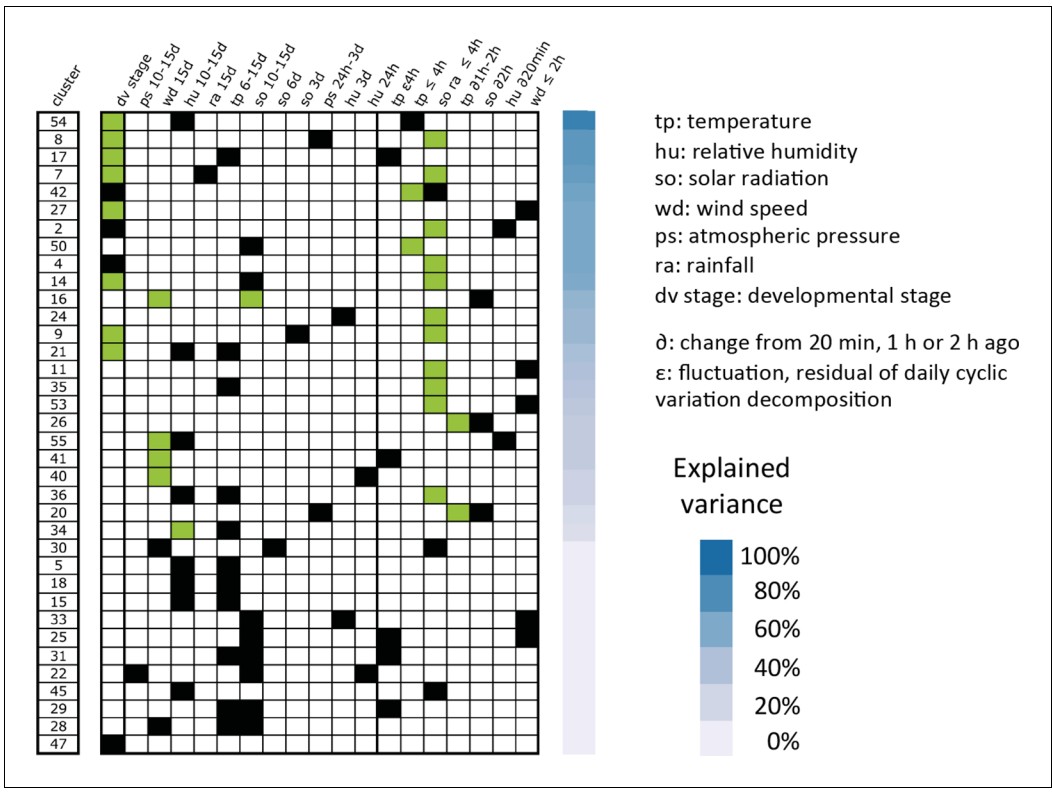

**Figure 7.** Transferability of two seasons irrigated field models to an independent dataset of rice gene expression under temperate climate. Models were selected for 60 clusters using our expression data in the irrigated field during the wet and dry seasons. The parameters from these models that could be transferred to explain the mean expression of the cluster mean in the PND are shown with a green box, the ones that were not transferred are indicated with a black box. The variance of the PND cluster mean explained by the partially transferred model is represented with decreasing shades of blue. PND, partial Nagano dataset.

was conserved in several PND cluster mean models: the 15-day average of wind speed, which was selected in four models, including one that explained 52% of the expression variance. These results show that a large part of climatic effects on gene expression vary with the type of climate; in particular, long-term effects.

As we found that several clusters enriched for genes involved in photosynthesis and development were among the most sensitive to the field context, we investigated whether this was also true for the seasonal and climatic context. We performed a GO term enrichment analysis on the clusters from the irrigated field analysis (*Supplementary file 1I*). We identified two clusters highly enriched for genes related to photosynthesis ($p<10^{-27}$ and $10^{-26}$, respectively), which were both modeled with season-dependent models. This result indicates that the transcriptional response of groups of co-expressed genes involved in photosynthesis is affected by the seasonal context. Two clusters (13 and 25) were enriched for genes associated with the developmental process and cell cycle. Cluster 13 was modeled with a season-dependent model while cluster 25 could be modeled with the same equation for both seasons of our experiment but no parameter of this equation could be transferred to model expression in the PND. This shows that the transcriptional regulation of some genes involved in development is not only sensitive to the field context but also to the season and climate type context.

To compare our method with the method from *Nagano et al. (2012)* with respect to the detection of climatic effects, we conducted another analysis of the PND using our model selection approach, determining clusters and models independently of the results obtained on our data (*Supplementary file 1H*). About a half of the models determined with the method by *Nagano et al. (2012)* do not have an environmental term and those that do, include the effect of a single climatic factor per gene, with a term that can be non-linear with diurnal changes in sensitivity. This makes their models intrinsically incompatible for a systematic direct comparison with ours. We thus chose to focus on two sets of genes identified by each method as having a clear environmental response and no developmental effect, and then evaluate the results of the alternative method on these genes. First, we selected from the results of our analysis genes highly correlated with their respective cluster means (r > 0.8) and belonging to three clusters with a low error model (MSE < 0.25) that included neither a developmental stage term nor a climatic parameter strongly correlated with the developmental stage. Among these 390 genes, 156 (40%) had a *Nagano et al. (2012)* model that included an environmental term for the timepoints we considered (between 8:00 and 16:00). Second, we chose genes with *Nagano et al. (2012)* models, including no developmental or circadian terms and with a linear environmental term for either temperature or solar radiation (which have the most reproducible effects) that explained more than 50% of the expression variance. When looking at the results of our analysis for these 214 genes, we found that 155 (72%) of them were in a cluster with at least one environmental term and a model error lower than 0.35 ($R^2 > 0.7$). Our method was thus more efficient in matching the Nagano et al. models with a clear environmental term than the *Nagano et al. (2012)* method was in the reciprocal comparison. This indicates that our approach has a higher sensitivity to the effect of climatic factors on gene expression, in addition to its ability to detect the co-occurring effects of multiple environmental parameters.

## Discussion

### Natural conditions provide a systematic view of transcriptional integration of multiple environmental signals

We used a model selection approach to identify relationships between major variations in global gene expression and environmental conditions/developmental stage. Focusing our analysis on groups of genes showing consistent variation across two different genotypes in each of the two seasons, we determined that most of these representative expression patterns could be explained through the combined effects of several environmental parameters, related to distinct climatic/soil-related factors and/or on different time-scales.

Co-occuring abiotic stresses trigger complex responses that cannot be predicted from the effect of single stresses (*Prasch and Sonnewald, 2015*), especially at the level of transcriptional regulation. Our dry season experiment, which included a drought period, presented ideal conditions to understand how limited water availability is integrated with other climatic signals at the transcriptional

level in the field in comparison to controlled conditions. One common finding of studies on the combined effects of drought and heat is that a high number of genes are differentially expressed only by the combination of both stresses (*Johnson et al., 2014*; *Rizhsky et al., 2004*, *2002b*). However, these results might be due to the fact that the differential expression induced by individual stresses is under the threshold of significance, while the addition of both small effects passes the differential expression threshold. Part of the assumed specificity could therefore be an artifact of the analysis. Although we cannot make a direct comparison with our analysis, as we do not have conditions testing the effect of drought alone, we can contrast gene expression responses between irrigated and rainfed fields during the dry season to assess how drought impacts the response to other abiotic stimuli and test whether there is indeed a specific response to drought combined with short-term climatic signals. Excluding clusters from groups 3 and 4, the expression of which is impacted by non-drought related field effects in the wet season, the different effects induced by drought in the rainfed field for clusters modeled with terms for short-term averages of temperature/solar radiation in the irrigated field during the dry season were (i) show the same response to short-term temperature/solar radiation with no or little shift in expression (clusters 18, 21 and 24); (ii) show an additive effect of drought over the short-term climatic response (clusters 4, 9, 12 and 30); or (iii) not display the short-term climatic response under drought (clusters 1, 8, 10 and 11). Only cluster 44, containing 204 genes, responded to short-term temperature/solar radiation variations in the dry season, specifically in the rainfed field. Our results therefore show that the main behaviors under combined drought and short-term abiotic signals are an addition of both responses and response to either one of the stimuli (often integrated with a long-term climatic response), while responses specific to the combination of both stresses are rare.

Genes that have been previously characterized for their role in a specific type of abiotic stimulus – light, temperature or water availability – and with a replicable expression pattern in our experiment generally displayed a transcriptional response to that stimulus. In many cases, these genes also seemed to respond to other abiotic stimuli, demonstrating the extensive connections between environmental pathways occurring in complex natural conditions. Some genes did not, however, have a pattern of expression consistent with what was measured in previous laboratory studies. For example, *OsCDPK7* and *OsMIOX* have been shown to be induced by drought (*Duan et al., 2012*; *Wan et al., 2007*). Nevertheless, in our data, there were no differences in mean expression between the irrigated and rainfed fields in the dry season for *OsCDPK7*, while *OsMIOX* was less expressed in the rainfed field than in the irrigated field during both seasons. These discrepancies may be explained by the fact that abiotic stresses are generally analyzed individually. Some genes induced by drought, when water availability is the only varying environmental parameter, might be mostly responsive to temperature during a combination of drought and heat stress.

Another possible explanation is the difference in time-scale of our field study compared with most laboratory-based experiments. The effect of an abiotic stress is often analyzed within hours of the stress treatment; in our field experiment, we examined climatic variables that changed constantly and gradually over a month. The time-scale of the study is especially critical in the case of drought stress, because in natural conditions, it usually occurs over the course of days or weeks. More generally, covering longer time frames than laboratory studies allowed us to observe environmental effects interact with plant developmental processes in an integrated way (*Allahverdiyeva et al., 2015*), as well as examine the effects of long-term seasonal climatic effects. Our results thus show that measuring gene expression under natural conditions and for long periods of time can help better assess the effect of multiple abiotic stimuli on gene expression.

## Field and climatic contexts had extensive effects on the environmental response of gene expression

In this study, we assessed the impact of different seasons, field environments and types of climate on weather-driven patterns of gene expression. Most of the seasonal differences were limited to the rainfed field and were related to the prolonged period, with very little precipitation occurring during the dry season. When analyzing only the irrigated field data, we found that 44 out 60 clusters (representing 72% of the genes in all clusters) were modeled with a common equation for both seasons (*Supplementary file 1G*). A fair level of reproducibility was also observed between different years of irrigated field culture for rice (*Nagano et al., 2012*).

In contrast, the type of field in which rice was cultivated had a major effect on the response to climatic conditions of many genes. A large part of the differences in climatic response between field environments that we detected were especially pronounced in the dry season and linked to the water status of the rainfed field. Some expression differences, however, seemed independent of this drought effect, as they were detectable in the wet season.

Finally, we used independent expression data to assess the effect of a different type of climate on transcriptional responses in irrigated conditions. We found that only a small subset of the gene expression patterns observed under the tropical climate of our experiment could be generalized to the temperate conditions of the *Nagano et al. (2012)* study. This result shows that we can expect to reproduce findings about gene expression despite seasonal variations, but we should be careful about generalizing results to other climates or soil conditions than the ones their study was set in.

When investigating the reproducibility of transcriptional responses, we found that gene expression patterns driven by short-term averages of solar radiation and temperature, and to a lesser extent long-term wind speed, were the most consistent responses across seasons, fields and climates. Short-term averages of solar radiation and temperature belong to the group of parameters that was used to model most of the 27 gene co-expression clusters, with the highest correlation between genotypes in our two-season analysis. In particular, the exponentially transformed short-term averages (15 min, 1 or 4 hr) of solar radiation (NL− ) that model stronger effects of variations in the lowest range of irradiance were selected to model at least one entire season or field subset for 12 of the 27 gene clusters of our analysis and could systematically be transferred from the models of our irrigated field data to explain the expression variation of the same clusters of genes observed under the Japanese temperate climate (*Supplementary file 1G*). In the conditions of the *Nagano et al. (2012)* experiment, short-term solar radiation variation was weakly correlated with variation in temperature and precipitation, showing that this effect is likely to be specific to irradiance level. The reason why variation in the lowest range of solar radiation values seem to have a stronger effect on gene expression (compared with higher irradiance levels) for some clusters needs to be investigated. One hypothesis is that responses to low light can rely mostly on changes in gene expression because they do not require as rapid a response as high light conditions (*Pfannschmidt and Yang, 2012*).

In contrast, we found that the effect of longer-term variation (10-, 15-day averages) in temperature and solar radiation were much less reproducible across seasons, fields and climates than short-term effects. One explanation is that, as these responses may be integrated over long periods of time, there is greater opportunity for them to be modulated by broad climatic and developmental factors. The only long-term environmental factor that could be transferred to model gene expression under the Japanese climate for several clusters was the 15-day average of wind speed. In our two-season analysis, it was selected to model across both seasons and fields cluster 32, which is enriched for thigmotropism genes. While there have already been studies aiming at identifying genes responsive to mechanical cues (*Lee et al., 2005*), they were limited to hour-scale laboratory experiments. Our results show that, in nature, the effect of wind on gene expression might be more prevalent on a week-scale, which is consistent with the long-term accommodation to repetitive wind loads observed in poplar (*Martin et al., 2010*). Further study of the genes in cluster 32 should contribute to the understanding of gene regulation in response to wind necessary to unravel the cellular processes involved in the time integration of mechanosensing (*Moulia et al., 2011*).

## Dynamic environmental stimuli are continuously integrated for the transcriptional regulation of development and photosynthesis

The analysis for GO annotation enrichment of the 27 clusters with the most reproducible expression patterns revealed functionally related groups of genes that are co-expressed in response to environmental and developmental signals. These results point to the biological processes most critical for plant physiological response to dynamic environments. The GO terms for biological processes with the most significant enrichment were nucleic acid metabolic process (and its parent term nucleobase-containing compound metabolic process) and photosynthesis.

Nucleic acid metabolic process related genes were highly enriched in clusters 4 and 6 ($p<10^{-18}$ and $10^{-22}$, respectively), which were also both enriched for developmental process and cell cycle associated genes. The strong enrichment in nucleic acid metabolic process related genes highlights the importance of a tight transcriptional regulation of this molecular process, probably linked to the

control of cell division, in the adaptation of the developmental program to environmental fluctuations (*Rymen and Sugimoto, 2012*). Both clusters had a negative correlation between field expression profiles in the dry season, while the field correlation was above 0.75 in the wet season, during which both cluster means were modeled with a positive term for the developmental stage parameter. This indicates an extensive impact of drought on the expression pattern of the genes in these clusters. As cluster 6 was modeled with several short-term environmental parameters, our results show that the transcriptional regulation of cell division-related aspects of growth integrates with the developmental program several time-scales of environmental stimuli.

Clusters 1, 5 and 13 were greatly enriched for photosynthesis related genes ($p<10^{-17}$, $10^{-9}$ and $10^{-19}$, respectively). These gene clusters were all down-regulated in response to drought, in accordance with previous results in *A. thaliana* (*Chaves et al., 2009*). Clusters 1, 5 and 13 were also modeled with short-term climatic parameters. These effects were detected mostly for the irrigated conditions, with positive terms for solar radiation in the case of clusters 1 and 5. In contrast, cluster 13 was modeled with a negative term for short-term temperature in both fields. These differences indicate that different components of the photosynthetic machinery might need to respond to different environmental stimuli. Although it has been thoroughly demonstrated that a large part of environmental effects on photosynthesis are regulated at the physiological and biochemical levels (*Kaiser et al., 2015*), the fine regulation of transcript abundance for these photosynthesis genes shown here confirms that the control of gene expression is also an important component of the modulation of photosynthetic activity (*Pfannschmidt and Yang, 2012*).

Focusing on our irrigated field data and the data from an independent field experiment (*Nagano et al., 2012*), we found that the expression patterns of groups of co-expressed genes involved in photosynthesis were affected by the seasonal context (independently of drought) and the environmental response of groups of genes associated with development was dependent on the climatic context. This suggests that the transcriptional regulation of both photosynthesis and development relies on complex mechanisms, resulting in the integration of numerous layers of environmental cues, short-term climatic fluctuations as well as steadier aspects of the plant surroundings.

## Conclusion

Understanding the dynamics of gene expression is a major challenge in biology. This is particularly difficult in the context of complex environments found in nature, as it requires unraveling the concurrent effects of multiple, fluctuating environmental signals on transcriptional patterns. Nevertheless, the ability to establish the environmental response of whole transcriptomes can have wide applications, including controlling engineered gene circuits (*Uhlendorf et al., 2012*) and predicting gene patterns in untested conditions (*Bonneau et al., 2007*; *Danziger et al., 2014*; *Nagano et al., 2012*). Our results suggest that while field environments can result in complex responses, one can nevertheless identify co-expressed gene clusters the mean expression of which can be accurately modeled with climatic, field, seasonal and developmental factors. Further work can integrate such models with other approaches, including gene network inference (*Bonneau et al., 2007*), genotype-by-environment interactions (*Marais et al., 2013*) and phenotypic modeling (*Aikawa et al., 2010*; *Satake et al., 2013*), to provide a more comprehensive picture of plant responses in natural environments. This, in turn, can be used in the design of improved crops (*Hammer et al., 2006*; *Mochida et al., 2015*) and the prediction of the ecological effects of climate change (*Stafford et al., 2013*).

## Materials and methods

### Plant growth, sampling and measurement

We used rice landraces for which seeds were available at the IRRI in Los Baños, Philippines, and that we knew are traditionally used for either lowland or upland cultivation. Azucena and Palawan are Filipino upland landraces while Pandan Wangi is a lowland Indonesian landrace. Azucena and Padan Wangi were grown in the dry season, while in the wet season we grew Azucena and Palawan.

The rice plants were grown at IRRI in a $25 \times 90$ m field divided into equal size rainfed and irrigated sub-fields. The dry season and wet season experiments took place during the months of January–February 2013 and July–August 2013, respectively. The two seasons differed for the amount and

frequency of precipitation as well as for temperature, which is generally lower during the dry season. During our experiment, the mean daytime temperature was 26.8°C for the dry season and 27.6°C for the wet season. The total precipitation during the experiment period was 84.5 mm in the dry season and 427 mm in the wet season. Precipitation of more than 1 mm was observed for 6 out of the 29 days of the dry season experiment and 22 out of the 29 days of the wet season experiment (*Supplementary file 2*).

In the irrigated field, one 21 day-old seedling was transplanted per hill with a spacing of 20 × 30 cm. For the rainfed field, direct seeding was performed with four seeds per hole, with a spacing of 20 × 30 cm; seedlings were thinned out to one per hill after 2 weeks. The composition of the fertilizer used was 120:30:30 nitrogen/phosphorus/potassium (NPK); it was applied as recommended. Carbofuran insecticide/nematicide was applied at 1 and 15 days after sowing (DAS) and herbicide was applied at 1 DAS in the upland ecosystem. Manual weeding and general plant protection were performed as needed.

The first sampling took place 16 days after transplanting seedlings in the irrigated field in the dry season and 23 days after transplanting in the wet season. Each sample consisted of six young leaves (of approximately the same size throughout the experiment) from six individual plants. Each leaf was immediately frozen in liquid nitrogen upon collection. We tried to reduce as much as possible the effect of circadian variation on gene expression, first between sampling time-points, by always starting the collection exactly 4 hours after sunrise. Second, to avoid a shift in expression within a sampling time-point due to the delay between the first and last collected samples, we ensured fast sampling by marking beforehand each plant and leaf to be collected. Collection took, on average, 13 min (between 11 and 15 min, except for the first time-point of the dry season, which took 18 min).

Averages of temperature, relative humidity, wind speed, solar radiation, precipitation and atmospheric pressure at the field site were recorded every 15 min using Wireless Vantage Pro2 ISS with 24-hr fan aspirated radiation shield from Davis Instruments (Hayward, CA). Measurements started 15 days before the first sampling day. After the ninth sampling of the dry season and during all of the wet season, from 1 hr before sampling to the end of sampling, we recorded climatic averages every minute. There was a weather station in the irrigated and rainfed fields, one of which also included a solar radiation sensor. No significant difference in measurements was detected between the fields, so the data of the station with the solar radiation sensor was used. We used 60 cm long 2710ARL tensiometers from Soilmoisture Equipment Corp. (Goleta, CA) placed 30 and 15 cm deep in the soil at four different locations, two per replicate of the rainfed field.

Plant height and tiller number were measured every 6 days for the same plants all along each season until the end of sampling (four plants per replicate in the dry season, twelve plants per replicate in the wet season).

## RNA sequencing

Frozen leaf tissue was ground manually with a mortar and pestle cooled in liquid nitrogen. Total RNA was extracted from about 200–400 µl of ground tissue using the RNeasy Plant Mini Kit (Qiagen, Venlo, The Netherlands), following the manufacturer's protocol and eluting the RNA in 40 µl of water. RNA was treated with Baseline-ZERO DNase (Epicentre, Madison, Wisconsin) according to the manufacturer's instructions then cleaned up with the Qiagen RNeasy Mini Kit and eluted in 32 µl of water. We assessed RNA quality using nanodrop and electrophoresis on an agarose gel. Total RNA, 4 µg, were depleted of ribosomal RNA using Epicentre Ribo-Zero Magnetic Kit for plant leaf tissue. We purified the depleted RNA with the Agencourt RNAClean XP kit (Beckman Coulter, Brea, CA). We constructed RNA libraries using the Epicentre ScriptSeq v2 RNA-Seq Library Preparation Kit, purifying the complementary DNA and libraries with the Agencourt AMPure XP System. We added barcode index using the Epicentre ScriptSeq Index PCR Primers. We quantified the libraries by Qubit (Life Technologies, Norwalk, Connecticut), with the DNA HS kit. Libraries quality and average fragment size was determined using the 2100 Bioanalyzer (Agilent, Santa Clara, CA) with high sensitivity DNA reagents and DNA chip. We quantified the libraries on the LightCycler 480 (Roche, Nutley, NJ) using the KAPA (Wilmongton, MA) Library Quantification Kit. Libraries were sequenced using HiSeq 2000 (Illumina, San Diego, CA) 51 bp paired-end sequencing, with either 6 or 8 libraries per lane. Each sample provided a mean of 58 million sequencing reads.

## Expression data analysis

The reads were aligned to the *O. sativa* Nipponbare – release 7 of the MSU Rice Genome Annotation Project reference genome (*Kawahara et al., 2013*) – 373,245,519 base pairs of non-overlapping rice genome sequence from the 12 rice chromosomes. Also included are the sequences for chloroplast (134,525 bp), mitochondrion (490,520 bp), Syngenta pseudomolecule (592,136 bp), and the unanchored BAC pseudomolecule (633,585 bp). The annotation contains 56,143 genes (loci), of which 6457 had additional alternative splicing isoforms resulting in a total of 66,495 transcripts.

We used Tophat (*Kim et al., 2013*; *Trapnell et al., 2009*) version 2.0.6 to align the reads, discarding low-quality alignments (quality score below 1). To count the number of reads that uniquely mapped to genes we used HTSeq (*Anders et al., 2015*) version 0.6.1. We compensated for variable sequencing depth between samples using the median-of-ratios method of DESeq2 (*Love et al., 2014*) version 1.6.3, and further performed a variance stabilizing transformation provided by the same package. We used the normalized count data for downstream analysis. This data is available in the Gene Expression Omnibus database (http://www.ncbi.nlm.nih.gov/geo/) under the accession number GSE73609.

We conducted a multi dimensional scaling of the normalized expression data where the samples clustered by genotype and field and to a lesser extent, season. We used these results to detect potential sample mislabeling and identified one sample switch. We also found that the sample for the first replicate of the Pandan Wangi rainfed field in the dry season, sixth time-point, did not cluster with any of the genotype/field groups so we removed it from the analysis and replaced it with a duplication of the second replicate.

We considered each subset of the data that consisted of one genotype and one season and excluded from the normalized expression dataset genes for which we detected no read for more than 40 samples in any of these subsets, which reduced the dataset to 22,144 expressed genes. We transformed the obtained value with the following function: $log2(x + 1)$, as to keep positive values of expression and averaged the biological replicates. We calculated the coefficient of variation of the log-transformed expression over the 240 data points for each expressed gene. We identified 1251 genes with a coefficient of variation below 0.01, which were considered as having a stable expression in our experiment and were removed from the analysis. A preliminary cluster analysis showed that genes with a very low expression level had a weak correlation with the center of their clusters, thus 2962 genes with a mean lower than 1 were not included in the clustering step. To remove the absolute differences in expression level between genotypes and seasons, which we did not intend to model, each genotype/season subset was centered.

## Clustering and gene ontology enrichment analysis

For each of the analyses, the expression data for each gene was scaled over the whole profile. Our clustering method was the Partitioning Around Medoids algorithm from the "cluster" package version 1.15.2 (*Maechler et al., 2015*) in R (*R development Core Team, 2011*) using $1 - r$, with r the Pearson correlation coefficient, as the distance between expression profiles. To choose the number of clusters (k) we first used several clustering indices but they gave inconsistent results and usually indicated a k lower than 20. To avoid running the risk of under-clustering, which would have resulted in averaging genes responding to very different ED factors and therefore losing transcriptional signal, we chose to constrain k only on the fact that most clusters should represent a major trend in transcriptional variation. We therefore chose k as the highest number of clusters for which no more than 5% of all the genes in the analysis belonged to "non-representative" small clusters, defined as containing less than 1% of all the genes in the analysis. The gene co-expression clusters were ordered according to the number of genes they contained so that cluster 1 was the biggest cluster.

Two sets of gene annotations were obtained from Gene Ontology (http://geneontology.org/page/about) from the November 2014 release. For the first one, the annotation database was queried via MSU locus identifiers; for the second one, the database was queried via Uniprot Ids, obtained via a mapping from MSU Id to OMA Id, and then OMA Id to Uniprot Id (mapping files available in the current release of OMA). A third set of annotations was obtained directly from OMA (http://omabrowser.org/oma/about/). All three annotation sets were then combined non-redundantly in order to produce the final annotation file for rice genes. The enrichment analysis was conducted using the GOstats package in R (*Falcon and Gentleman, 2007*).

## Model selection

To represent each cluster, we calculated the average scaled expression of all the genes in the cluster. This "cluster mean" was used as the output data in our modeling approach. Climatic, field and developmental parameters constituted the input data. We calculated averages of temperature, relative humidity, wind speed, solar radiation, precipitation and pressure for 15 min 1, 4, and 24 hr; and 3, 6, 10 and 15 days before sampling. Short-term changes in temperature, humidity, wind speed and solar radiation were determined by calculating the difference between the value of each measurement at the sampling time and 20 min ($\delta$20 min), 1 hr ($\delta$1 hr) and 2 hr ($\delta$2 hr) earlier, using 5, 10, and 30 min averages, respectively. We evaluated temperature fluctuations by decomposing daily variation in temperature with the seasonal decomposition by loess (stl) function in R and calculating 1 hr ($\varepsilon$1 hr), 4 hr ($\varepsilon$4 hr) and 24 hr ($\varepsilon$24 hr) before sampling averages of the remainder of the decomposition. The value used for the soil moisture parameters was a mean of measurements from four tensiometers, two in each replicate of the rainfed field. Exponentially transformed values of solar radiation used the following equations:

$$NL+ (x) = exp((x-400)/200)$$
$$NL- (x) = exp((400-x)/200)$$

We designed a parameter that would give an estimate of the developmental stage, rather than age, of the plant to be able to compare appropriately the plants in the rainfed and irrigated fields as they followed different developmental itineraries relative to their age in days. This parameter used three stages as fixed points: transplanting stage (given a value of 0), corresponding to the actual transplanting event for the irrigated field and determined for the rainfed plants as the stage where they were the same height as just transplanted irrigated plants, end of tillering production (40) and heading stage (100). Intermediary time-points were calculated linearly between these fixed points. Input parameters were centered per genotype and season and scaled over the whole dataset to match the expression data.

When input parameters were more highly correlated than the highest genotype correlation of all clusters (r = 0.98), the correlated parameters were averaged together, as we would not have enough precision in the expression data to discriminate between them.

The same method was used on each cluster of a given set to select a model. It is described here for the analysis of one genotype in two seasons *Source code 1*; the analysis of two genotypes in one season is identical and the analysis of the irrigated field with averaged genotype follows the same principle.

1. We used the high domensional inference (hdi ) function of the hdi R package (*Meinshausen et al., 2009*; *Meinshausen and Bühlmann, 2010*) using the lasso algorithm for the parameter selection, with the "stability" method, B = 300, EV = 2, threshold = 0.65 and fraction = 0.85, to select a stable subset of all the input parameters using the whole 60 data points.
2. All possible combinations of one, two and three parameters from this subset were used to calculate linear regression models, the MSEs of which were computed using five times five-fold cross-validation. The models were compared using the BIC to select the optimal model that fits the data while limiting overfitting. To avoid linear models containing correlated parameters in the same equation, which does not bring much more information than only one and increases the risk of overfitting the data, when two parameters from the subset selected with the hdi function were correlated with r > 0.85, we only kept in the group of tested parameters the one that resulted in the best model. The linear model selection step was applied to select a model for the 60 data points together, as well as for pieces of the cluster mean: the 30 data points of the dry season alone, wet season, irrigated field, rainfed field, as well as the 15 data points of the irrigated field in the dry season, rainfed field in the dry season, irrigated field in the wet season and rainfed field in the wet season.
3. We used these results to form seven piecewise models to be compared to the model selected for the whole dataset (*Figure 1B*):
    1. model in one piece (calculated with the 60 data points)
    2. dry season + wet season
    3. irrigated field + rainfed field
    4. dry season + irrigated field in the wet season + rainfed field in the wet season

5. wet season + irrigated field in the dry season + rainfed field in the dry season
6. irrigated field + rainfed field in the dry season + rainfed field in the wet season
7. rainfed field + irrigated field in the dry season + irrigated field in the wet season
8. irrigated field in the dry season + rainfed field in the dry season + irrigated field in the wet season + rainfed field in the wet season

We chose from these eight models using the BIC. The MSE of composite models was computed by assembling the squared residuals from cross-validation calculated from each distinct linear equation individually into one vector of squared residuals for the whole model and the number of parameters per model was the sum of the number of parameters of each equation.

4. In the case of strong differences in ED responses between fields and seasons, the hdi function might not select ED parameters that can fit each field or season using as output the expression data from both seasons and fields. To take that possibility into account, we repeated the hdi parameter selection with season and field subsets of the data, with B = 300, EV = 2, threshold = 0.7 and fraction = 0.7. We repeated the model selection steps described above using first a pool of the parameters selected using the dry and wet season subsets and then a pool of the parameters selected with the rainfed field subset and the irrigated field subset. We used the BIC to choose from the three models selected from these three groups of parameters.

## Analysis of the data from *Nagano et al. (2012)*

We used microarray data from two studies (*Nagano et al., 2012*; *Sato et al., 2011*) available on the GEO website (accession numbers: GSE36040 and GSE21494), that was already normalized and log-transformed. We used only data from sampling points collected at 8:00, 10:00, 12:00, 14:00 and 16:00 as these times were far enough from sunrise (around 4:30) and sunset (around 19:00). We only used data from samples collected before flowering, from 15 to 62 days after transplanting. Sunrise time varied little enough during that period (from 4:23 to 4:34) that it could be estimated that a series of samples collected at the same time of day would have negligible variation in circadian clock. There were eight samples collected at 8:00, 12 samples at 10:00, eight samples for single replicates and 24 samples for triplicates of eight sampling points at 12:00, eight samples at 14:00 and eight samples at 16:00. We excluded from the analysis genes with an expression value below −7 in more than 17 samples, keeping 19,837 genes, 16,659 of which overlap with the ones in our analysis. We averaged the biological triplicates, resulting in a total of 52 data points. For every gene, we centered individually each of the separate profiles corresponding to a given time of day, thus eliminating potential circadian clock driven differences between these times of sampling. This data constituted the PND.

We tested the transferability of the models determined with the irrigated field data of our experiment, limiting our evaluation to models that were season independent and explained more than half of the variance of the cluster mean. Using the same gene distribution as our irrigated field clusters, we calculated cluster means for the PND. We calculated climatic parameters in the same way as we did for our experiment. The developmental stage parameter was the number of days after transplanting. Input parameters were centered per time of day to match the expression data. We applied the second step of our model selection method to these profiles to select models common to all five time-of-day profiles, using as a subset of parameters the ones in the model determined for our irrigated field data instead of the hdi pre-selection. We only kept a parameter in the new model if it was fitted with a coefficient of the same sign as in the original model.

For the independent analysis of the PND, we used the same method as for our data, which produced 60 gene clusters. The model selection method was a simplified version of the one used for our data. We added to the set of input parameters short-term averages of 8 and 12 hr to account for a possible longer effect of same day conditions, as some samples were collected later in the day than in our experiment. We only selected models common to all five time-of-day profiles, using a unique set of parameters selected from the cluster mean in its entirety and choosing from all possible linear regressions with no more than three parameters using the BIC.

We used the Field Transcriptome Database in *Oryza sativa* (http://fitdb.dna.affrc.go.jp/) to identify the genes showing a clear environmental response detected by the Nagano et al. models, choosing the ones that had models where both the $R^2$ of the overall model and the $R^2_d$ of the environmental parameter were over 0.5, with no developmental or circadian terms and an

environmental term for either temperature or solar radiation with no gate or a sinusoidal gate and a dose-dependent response.

## Acknowledgments

We thank Reynaldo Manuel Jr. for his help with plant growth measurements and collecting the leaf samples, Eloisa Suiton, Julius Borgonia, Godofredo Perez, Josefina Mendoza, Paul Cornelio Maturan, Carlos Casal Jr., and Danilo De Ocampo for their help in the field and the staff of IRRI's Climate Unit for their help with climate data. We are grateful to Noah Youngs for gathering the GO annotations. We also thank Atsushi Nagano for providing the weather data, detailed result tables and additional details concerning the work published in Naganoet al. (2012). This work was funded by a grant from the National Science Foundation Plant Genome Research Program.

## Additional information

### Funding

| Funder | Grant reference number | Author |
| --- | --- | --- |
| National Science Foundation | IOS-1126971 | Michael Purugganan |

The funders had no role in study design, data collection and interpretation, or the decision to submit the work for publication.

### Author contributions

AP, Conception and design, Acquisition of data, Analysis and interpretation of data, Drafting or revising the article; CH, OW, Analysis and interpretation of data, Drafting or revising the article; ZJG, IP, CM, Acquisition of data, Drafting or revising the article; RSM, Acquisition of data, Contributed unpublished essential data or reagents; EMS, MP, Conception and design, Drafting or revising the article; RB, Conception and design, Analysis and interpretation of data, Drafting or revising the article

## Additional files

### Supplementary files

• Supplementary file 1. Detailed characteristics and models for all the clusters in the two-season analysis; results used for the comparison of the partial Nagano dataset with our analysis.

• Supplementary file 2. Weather data for the dry and wet season experiments.

• Source code 1. Clustering, model selection for the two-season analysis and analysis of the partial Nagano dataset.

### Major datasets

The following datasets were generated:

| Author(s) | Year | Dataset title | Dataset ID and/or URL | Database, license, and accessibility information |
| --- | --- | --- | --- | --- |
| Plessis A, Hafemeister C, Bonneau R, Purugganan M | 2015 | Rice gene expression in irrigated and rainfed fields during two seasons | http://www.ncbi.nlm.nih.gov/geo/query/acc.cgi?acc=GSE73609 | Publicly available at the NCBI Gene Expression Omnibus (Accession no: GSE73609). |

The following previously published datasets were used:

| Author(s) | Year | Dataset title | Dataset ID and/or URL | Database, license, and accessibility information |
|---|---|---|---|---|
| Sato Y, Antonio B, Namiki N, Nagamura Y | 2012 | Diurnal and circadian gene expression profile of leaf throughout the entire growth of rice in the field | http://www.ncbi.nlm.nih.gov/geo/query/acc.cgi?acc=GSE36040 | Publicly available at the NCBI Gene Expression Omnibus (Accession no: GSE36040 |
| Yutaka Sato | 2011 | Transcriptomic analysis of rice | http://www.ncbi.nlm.nih.gov/geo/query/acc.cgi?acc=GSE21494 | Publicly available at the NCBI Gene Expression Omnibus (Accession no: GSE21494). |

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
