## [Decision Letter]

Thank you for submitting your work entitled "Multiple abiotic stimuli interact to regulate rice gene expression under field conditions" for peer review at *eLife*. Your submission has been favorably evaluated by Detlef Weigel (Senior editor) and three reviewers, one of whom is a member of our Board of Reviewing Editors (Daniel J Kliebenstein).

The reviewers have discussed the reviews with one another and the Reviewing editor has drafted this decision to help you prepare a revised submission.

Summary:

This work begins a new effort to combine modern genomics methodologies with new computational approaches to look at how organisms exist within the environment. This includes an effort to incorporate all of the fluctuations in this environment in an attempt to better parse the transcriptomic response of the organism to this highly changing system.

Overall, this manuscript is highly interesting and begins to extend the genomics effort into more real world settings.

Essential revisions:

There were two fundamental concerns:

1) The first was that it was felt that the new computational approaches should be tested against another independent data set. Specifically it is suggested that the authors create models with their data on the parameter shared with Nagano et al., since the claim is that they have a new pipeline for data analysis/model building. Then, they should test these models with the data from Nagano et al. to demonstrate the added value.

2) Secondly, the writing could use further improvement to help convey to both the biologists and the computational scientists what the inherent novelty of this study is to both groups. Alternatively, it might be suggested to focus solely on one group and provide them with insights that pertain to that group.

Reviewer #1:

My major concern was that it was difficult to parse out what was or was not truly novel in the results and analysis. I understand that these articles are splitting a divide between the computational and the biological communities each with their own and often contrasting expertise. As such, I would suggest an editorial reworking to make the central messages to be as simple and direct as possible.

Reviewer #2:

The main goal of this study is to determine the effects on multiple environmental factors/components on gene expression in natural field conditions for rice. The claim is that computational approach taken after the gathering and assembling the transcriptome profiles (based on RNA-seq) for three rice landraces with two different systems of cultivation (two biological replicates per field type – rainfed and irrigated) in two phases of cultivation (dry and wet), allows establishing conclusions about interacting abiotic stimuli in field environments. The data set consists of expression levels for ~22,300 genes (after preprocessing, to remove invariable transcript profiles and transcripts not detected in majority of samples). In addition, the authors attempted to test the validity/robustness of the proposed approaches on an independent data set.

In the following, I will only focus on detailed review of the shortcoming of the statistical modeling strategy proposed and employed in this study. The workflow is summarized as follows: (1) conduct clustering of transcript profiles, (2) pre-select environmental parameters, (3) build models with all combinations of one and two of the preselected parameters on different partitions of the data from each cluster (e.g., type of field, season, and combinations thereof); here, cluster means were used as responses and (transformations of) environmental factors were treated as predictors, (4) select models based on Bayesian information criterion (BIC) taking into account model complexity, (5) combine selected models, and (6) inspect the congruence of models (with respect to their coefficients) in different partitions to draw conclusions about effects of climatic fluctuations. In addition, enrichment analysis for different functions in the determined clusters was used to provide biological interpretation.

There are numerous technical issues and difficulties with the chosen strategy, some of which could be remedied, and other which are to be deeply questioned. These are present in every step of the workflow, which led me to question the interpretation and conclusions drawn.

For instance, in step (1), no validation for selection of the number of clusters is provided, nor it is assessed how this may affect the results; the authors could use various cluster quality indices to back up their claims; at this point, it is also not clear which distance measure was used in the PAM clustering approach. It is also advisable that the same centering and scaling strategy is used from the beginning of the workflow, so that consistent results are expected.

In step (2), the set of parameters used was first pruned, by arbitrarily removing highly correlated parameters; there are many ways in which this could be done – was the parameter with largest number of high correlations removed first, or was another strategy used? How would this affect the final results obtained and the subsequent biological interpretations? Most critically, the authors only state that the hdi package in R was used, without indicating if they used it with the default setting (lasso) for the variable/parameter selection. At this stage, I do not know if the parameters were selected based on p-values (from cross-validation), since lasso has multiple solutions, which again would affect the preselected parameters! Moreover, centering and scaling was here done on the entire data set, which will certainly affect the findings should this be done differently (e.g., on a subset of the same data, as the authors do later on).

The same lack of robustness checks is present in step (3), where the pre-selection was repeated for step 2, simply because some of the selected models failed the MSE test (< 0.15).

In step (4) it is advisable to determine MSE from cross-validation, and to only use BIC. I do not understand why the combination of MSE and BIC should be used for the selection.

Finally, the combination of models from step (5) is nowhere detailed; are they only inspected for congruence of coefficients or are they summed to make predictions for the additive effects of two environmental stimuli?

In the enrichment analysis, why were the annotations found across the three annotation resources not combined in a weighted fashion, so that GO terms more often encountered are given a higher more weight? The simple combination employed may distort the findings, particularly for the under-representation of classes (if this was tested at all).

Finally, it would have been interesting to see how the models performed in the case of the data from Nagano et al.; rather than building models on the independent data (any strategy could have been used to this end, not only the heuristic suggested by the authors), the transferability of the models should have been assessed and commented on. This is in fact the most challenging part and would shed light on bridging the greenhouse-field gap.

At this stage, I also wonder why the profiles from different genes involved in particular process were not used to derive the models (this would have avoided the enrichment analysis as well as the clustering and might have made it for a more concrete story).

My conclusion is that the authors used lasso (without ever having stated it and in combination with questionable strategies to remove some of the parameters) followed by classical regression techniques to arrive at the conclusion that correlated responses (whatever they may represent) result in similar models – this is an expected finding and it does not address the main aim of the study. Should an existing variable selection method been used as the only strategy or if latent variables were used on their own with simple regression techniques the authors would have had a streamlined and feasible way to assess the uniqueness, quality, and robustness of the findings, which now is almost impossible to objectively assess.

Reviewer #3:

Plessis et al. is a field genomics paper in rice that examines the relationship between environmental variables across cultivars of rice across 2 seasons (one cultivar is represented in each season). The main goal of the paper is to develop a method to find environmental variables that are highly correlated with gene expression. The authors were careful in how the tissue for RNA-seq was collected so as to minimize confounding effects of time between samples and the circadian clock. Additionally, most of the code used to do the analysis was provided which made detailed thought about the method easier. The authors even used previously published rice data to apply their method to. In general, the paper would be of general interest to the readers of *eLife*.

I have three major criticisms of the paper in its current form:

1) This is a large dataset with many axes of variation with one of the main conclusions being "additive and interactive effects of distinct environmental conditions on gene expression are widespread". While simplicity is nice for interpretation ("In our effort to produce simple models, we limited the number of parameters per linear equation to two"), finding the best two parameters is an arbitrary cutoff. This needs to be justified statistically by either simulations or references. Furthermore, there is discussion of interactions when these simple models do not account for interactions between variables. It is unclear from the analysis and text how interactions are defined and thus interpreted.

2) The number of PAM clusters that were chosen is arbitrary and could influence the interpretation. There is no code provided for the PAM cluster analysis. Additionally, this is a concern because while the original data 240 number of samples were used in PAM clustering with 50 clusters, the revisited Nagano et al. 2012 data had only 52 RNA samples (that were averages across replicates) but 50 clusters was chosen for this data set as well. The risk here is over-fitting the Nagano data. In the same R package that was used to generate the clusters ("cluster"), there are functions (see clusGAP) to calculate the GAP statistic to determine the appropriate number of clusters given the dataset (Tibshirani et al. 2001).

3) In addition to the technical concerns above, as written there is not enough of a comparison between this modeling approach and the one of Nagano et al. 2012 to show what this new approach has to add with respect to predicting gene expression patterns in the field. I think that the paper could benefit greatly from showing how the model selected from simple to complex would compare to using a full model including interactions with all the terms and gradually dropping terms out to compare models like in the Nagano et al. 2012 paper (although Nagano did not deal with interactions). For example, model selection on: lm(gene expression ~ genotype*season*timepoint*treatment). Building on this, I also think there is a missed opportunity here to do some mixed effects modeling (see lme4 R package) that could take into account the correlation of gene expression within and between all the treatment, genotype time of year combinations.

[Editors' note: further revisions were requested prior to acceptance, as described below.]

Thank you for resubmitting your work entitled "Multiple abiotic stimuli are integrated in the regulation of rice gene expression under field conditions" for further consideration at *eLife*. Your revised article has been favorably evaluated by Detlef Weigel (Senior editor), a Reviewing editor, and two reviewers (Reviewer #2 and Reviewer #3). The manuscript has been improved but there are a few remaining issues that need to be addressed before acceptance, as outlined below:

The reviewers felt that the manuscript nicely shows that even though climate and local field environment have large effects on gene expression, their effects can be captured with relatively simple models. While there are not yet sufficient data to be predictive for climates or field environments that have not been evaluated yet, the finding suggests that a relatively limited amount of such data sets will enable predictive modeling. This should be emphasized in the Abstract, Introduction and Discussion.

Both reviewers also identified a few technical details that need more clarification before final acceptance. All reviewers and editors for this manuscript felt that there still needs to be significantly more effort to reach the biological reader and convey the novel insights. You state at the end of the Abstract: "We show that new insight can be gained from studying the effects of co-occurring abiotic stimuli in complex dynamic environments." Please state more clearly what these are. All efforts to clarify these insights for both a biological and quantitative readership will greatly improve the future impact of this manuscript on the field.

More detailed requests for clarification are found below.

Reviewer #2:

The authors have carefully considered and addressed the comments raised by the reviewers with respect to the selection of number of clusters (already a non-trivial task, given the number of different measures they considered), number of piecewise models (per BIC only), and averaging of profiles. However, the issue with the model simplification by pruning correlated parameters seems to differ between the response and the main text. More specifically, it is not clear how the decision in paragraph two, subheading “Modeling the effect of climatic factors on transcriptomic variation in different field environments” was implemented, as this can be carried out in many ways. This statement does not correspond to the response provided in your letter, whereby the correlation is set to 0.98 and the parameters were averaged (this strategy of averaging provides a deterministic and reproducible outcome).

In paragraph four, subheading “Modeling the effect of climatic factors on transcriptomic variation in different field environments”, it should be included that the BIC is calculated on the joint piecewise models; I suppose the number of parameters corresponds to the number of predictors summed over all models, or does it correspond to the number of unique predictors over all models? This will have effect on the final findings.

The reciprocal analysis of the partial Nagano data (PND) appears to have pulled out the environmental effects predictive for the transcriptomic changes, and I find that it has added some ideas about the difficulty of the problem of model generalizability.

I would suggest that the claim about "designing model selection approach" is toned done, as the authors themselves state in the response letter that "we do not claim to provide any notable advances in this paper for the computational community". While this may be an article aimed for the genomics-enabled biologist, it will be read by computational biologists interested in the large data resource provided.

Finally, while the authors provided a pipeline based on well-established methods, the interested reader may want to know what is the added value with respect to now classical approaches (e.g., mixed effect models), as one of the other reviewers already suggested.

Rephrasing of some key sentences may be needed, to fully match what was stated in the main text and response letter:

1) The statement in the Abstract "we show that new insights can be gained" should indicate what these insights precisely are.

2) The impact statement should be clarified – one can determine a model for any set of variables; however, the pressing problem is to show that the model explains a major part of the variance and can be used for predictive purposed. Therefore, it also needs rephrasing.

3) The statement of the major findings – "Our method allowed for the detection of additive effects of several environmental factors and differences in gene expression patterns between fields, genotypes or seasons" – is not strong enough, since any data analysis method for differential analysis does essentially the same.

Reviewer #3:

1) In re-reviewing this paper the authors have done a great deal to clarify the modeling approach and have provided many details that were requested. I think that they have made great improvements in the technical aspects of this paper. However, now that the modeling details are clearer the manuscript could be vastly improved for readability for a biological audience (as a stated goal in response to reviewer 1's comment) if the biological question was clearly outlined in the Introduction and followed through the Methods, Results and Discussion.

In the Abstract: "We show that new insight can be gained from studying the effects of co-occuring abiotic stimuli in complex dynamic environments."

Now that the technical aspects of the methods are clearer I am struggling to see what new biological insight is gained from this paper. My main criticism of this paper as a whole in this new version is that it reads as a new way to do exploratory data analysis in a large-scale field transcriptomics experiment, but is not framed to ask a direct biological question with the data/analysis.

In my last review I stated: "as written there is not enough of a comparison between this modeling approach and the one of Nagano et al. 2012 to show what this new approach has to add with respect to predicting gene expression patterns in the field." I followed this with some suggestions as ways to approach this using a mixed model approach that could be applied to genes or gene clusters of interest. The authors have done extra work to apply their method to the Nagano et al. 2012 paper, but I still do not see what new biological insight is gained from the cluster method over gene-wise predictions. If the truly novel aspect of this dataset (compared to Nagano et al. 2012) is the drought/non-drought comparison then could the paper focus much more on this comparison?

2) Interactions implied in Introduction and Conclusion, but not tested in models:

Introduction: “In addition to the dynamic nature of field conditions, the interaction between multiple stimuli is another major cause of discrepancies in plant responses/phenotypes between laboratory and field conditions.”

Conclusion: "By working in natural, complex conditions in the field, we can examine interactions between the effects of factors that vary at distinct time-scales, like temperature and water availability."

In the Authors’ response: "We agree that the use of the term "interaction" in the context of linear models can be confusing. We have therefore removed any mention of interactions in the interpretation and discussion of our modeling results to make our meaning clearer."

Although the authors disagreed with me about the including models with interaction terms for the entire dataset, I still think it would be a good approach to make distinct comparisons between genes/gene clusters and how they respond to the drought treatment.

For example, in the subsection “The agricultural field environment strongly affects transcriptional responses to climatic fluctuations”, the authors state: "This result suggests that, in our experiment, a group of abiotic stress response genes responded to a greater extent to high light/heat stress than drought stress, while another group of abiotic stress response was much more sensitive to the effect of drought." Why not subset the data based on this observation and test these comparisons directly with a follow up model that includes an interaction term?

3) Do these gene expression clusters mean anything for the plant as far as other phenotypes are concerned? There is a missed opportunity here to expand what is found in the clusters into a story about how the environment influences gene expression and how that gene expression might influence plant growth/development. The Discussion starts to get at this, but the message is lost in the details about what a few genes from each cluster mean. The clustering/model selection approach is a great way to reduce this complex dataset to fewer significant parameters but as this manuscript is written does not help in biological interpretation of the clusters. This is also where a clear biological goal of the modeling/method/experimental design is necessary and could improve this manuscript.

---

## [Author Response]

Essential revisions:

There were two fundamental concerns:

1) The first was that it was felt that the new computational approaches should be tested against another independent data set. Specifically it is suggested that the authors create models with their data on the parameter shared with Nagano et al., since the claim is that they have a new pipeline for data analysis/model building. Then, they should test these models with the data from Nagano et al. to demonstrate the added value.2) Secondly, the writing could use further improvement to help convey to both the biologists and the computational scientists what the inherent novelty of this study is to both groups. Alternatively, it might be suggested to focus solely on one group and provide them with insights that pertain to that group.

We appreciate the favorable evaluation of our work and the interest for our contribution to the new field of ecological systems biology. We found that we were able to address reviewer and editor comments, either through modification of the manuscript or through additional analysis. Although a detailed account of our revisions to the work and the manuscript follow, we briefly summarize the main points of revisions here:

1) We have used the comments and suggestions of the reviewers to modify our computational method and make it more consistent across datasets: we have set up a criterion to choose the number of clusters, revised our centering and scaling steps and simplified the model selection steps.

2) We have modified the comparison of our results with the ones from the Nagano et al. paper.

3) We have extensively edited the manuscript to address the concerns and required improvements of the reviewers. Specifically, we believe that our revised manuscript will now be clearer to biologists, our intended audience.

Reviewer #1:

My major concern was that it was difficult to parse out what was or was not truly novel in the results and analysis. I understand that these articles are splitting a divide between the computational and the biological communities each with their own and often contrasting expertise. As such, I would suggest an editorial reworking to make the central messages to be as simple and direct as possible.

The novelty in our work is to be found in the results that can be obtained from an approach combining focused experimental field measurements and a computational analysis that together were designed to specifically detect environmental effects on gene expression in fluctuating conditions. The computational analysis itself relies on well-established component statistical tools but contains a non-trivial amount of novelty via the combination of these modules into a working pipeline that identifies and models the environmental response of co-regulated groups. That said, we do not claim to provide any notable advances in this paper for the computational community, but rather aim the manuscript at genomics-enabled biologists with no extended statistical knowledge. As stated in the Introduction, environment-targeted transcriptomics studies carried out over several weeks or including more than two environmental perturbations are extremely rare and in this regard, our results pertaining to the combined effect of drought, temperature, solar radiation and wind on gene expression in the course of a month are truly original. Compared to the other main dataset of rice transcriptomics in the field (Nagano et al., 2012), ours expends the range of environmental conditions provided, with a rainfed field in addition to the irrigated field and two contrasted seasons, the dry season comprising a drought period. In the new version of our manuscript, we have highlighted further these various aspects of the novelty of our work.

Reviewer #2:

*For instance, in step (*1*), no validation for selection of the number of clusters is provided, nor it is assessed how this may affect the results; the authors could use various cluster quality indices to back up their claims; at this point, it is also not clear which distance measure was used in the PAM clustering approach. It is also advisable that the same centering and scaling strategy is used from the beginning of the workflow, so that consistent results are expected.*

Reviewer #3:

2) The number of PAM clusters that were chosen is arbitrary and could influence the interpretation. There is no code provided for the PAM cluster analysis. Additionally, this is a concern because while the original data 240 number of samples were used in PAM clustering with 50 clusters, the revisited Nagano et al. 2012 data had only 52 RNA samples (that were averages across replicates) but 50 clusters was chosen for this data set as well. The risk here is over-fitting the Nagano data. In the same R package that was used to generate the clusters ("cluster"), there are functions (see clusGAP) to calculate the GAP statistic to determine the appropriate number of clusters given the dataset (Tibshirani et al. 2001).

Due to the similarity of these two comments we address them in aggregate here.

It is an oversight on our part that we did not mention the distance used for the clustering (1 – Pearson correlation coefficient) and it has now been corrected. We also now provide the code for the clustering step in our Source Code file. According to the suggestion of reviewer 2, we have changed our scaling and centering strategy to make it more consistent. We were only interested in differences between genotypes related to their environmental responses and not in mean differences in expression level. Differences in mean expression level between seasons could be caused by differences in climatic factors as well as other factors that we did not include in our analysis for example day length or non-water related soil characteristics. To avoid attributing mean differences between seasons to the wrong effect, we preferred not to take into account these differences in mean expression level. For these reasons, each quarter of the dataset constituting one genotype in one season was centered, which removed differences in mean level of expression between genotypes and between seasons. This centering was performed once before the data was used in either the dry season only or two seasons analysis. Scaling was done once, for each gene over the whole dataset used in each analysis, prior to clustering. The exact same treatment is applied to the environmental parameters. We maintain that this is the correct pre-treatment to the data (and that this mitigates genotype/batch effects that would otherwise obscure environmental response signals in the data).

Modeling cluster means was a way to focus our analysis on major transcriptional variations that are more reproducible than the expression profiles of individual genes (paragraph two, subheading “Simple models can explain transcriptomic variation across two seasons”). Furthermore, our results show that clusters with a lower number of genes are more likely to have less reproducible expression profiles (measured with genotype correlation, [Supplementary-material SD1-data]). For this reason, we decided to limit the number of clusters to avoid small clusters that would not contain enough genes to be representative of a major trend in global gene expression and might not have a reproducible expression profile. To be more consistent across our different analyses, we have now specified a criterion for this limitation: we have constrained the number of clusters so that no more than 5% of all the genes in the analysis belong to “non-representative” small clusters, defined as containing less than 1% of all the genes in the analysis. To find an optimal number of clusters below this constraint for our main analysis (two-season analysis), we have used the GAP statistic, which had local maxima at 18, 23, 31, 38, 43 and 48 clusters. We then tried other indices, which also had local maxima: 8, 14, 28, 36 and 46 for the silhouette index, 9, 31, 36, 40, 42, 49 and 52 for the Dunn index, 14, 29 and 43 for the SD validity index. If we were to use as a number of cluster the first local maximum provided by one of the indices cited above, it would be between 8 and 18. We have considered the dangers of under-clustering (setting k too small) and over-clustering (setting k to be larger than the ‘true’ number of co-regulated groups of genes). Upon careful consideration we find it is clear that under-clustering has dire consequences to our downstream analysis (averaging together genes that are unrelated dampens/destroys signal) while over-clustering is of minor consequence (splitting a large cluster into two smaller clusters does not produce spurious results, but simply reduces the efficiency of the summarization afforded by the clustering). To avoid the risk of under-clustering, we therefore chose k as the highest number of clusters that would satisfy our requirement for representativeness of most clusters. This is now discussed more clearly in the Materials and methods section.

Reviewer #2:

*In step (*2*), the set of parameters used was first pruned, by arbitrarily removing highly correlated parameters; there are many ways in which this could be done – was the parameter with largest number of high correlations removed first, or was another strategy used? How would this affect the final results obtained and the subsequent biological interpretations?*

We changed our pruning strategy to avoid taking arbitrary decisions. We set our threshold of correlation to 0.98, the highest genotype correlation of all clusters, an indicator of the noise in the expression data we are modeling. When several parameters were correlated over this threshold, we averaged them to create a new parameter replacing them all.

Most critically, the authors only state that the hdi package in R was used, without indicating if they used it with the default setting (lasso) for the variable/parameter selection. At this stage, I do not know if the parameters were selected based on p-values (from cross-validation), since lasso has multiple solutions, which again would affect the preselected parameters! Moreover, centering and scaling was here done on the entire data set, which will certainly affect the findings should this be done differently (e.g., on a subset of the same data, as the authors do later on).

This information was not available in the Materials and methods section but could be found in the R code provided. This has been corrected and the Methods section expanded accordingly. We did use the lasso, with the stability method of the hdi function. Stability selection, as implemented in the hdi package, constructs many lasso paths, in our case 300 solutions based on a random 85% of the data each. It then looks at how frequent each parameter was in the first q parameters in the paths and selects only parameters with a frequency above a certain threshold. Variable q and the frequency threshold can be set using an upper bound for the number of false positives, in our case 2 (see Peter Buhlmann, Markus Kalisch, and Lukas Meier, 2014 for details).

At this step, we now only center the field subsets of the data (gene expression and modeling parameters alike) to avoid getting models with an intersect.

*The same lack of robustness checks is present in step (*3*), where the pre-selection was repeated for step 2, simply because some of the selected models failed the MSE test (< 0.15).*

This is a misunderstanding of our method: we have not been clear enough about this particular step in the Materials and methods section and this has now been corrected. We systematically repeat the pre-selection, independently of any test, and choose between the three models obtained with the three different pre-selections using the BIC.

*In step (*4*) it is advisable to determine MSE from cross-validation, and to only use BIC. I do not understand why the combination of MSE and BIC should be used for the selection.*

This is a good comment, and we agree that this could be improved. We have thus changed our model selection approach accordingly. We now use only the BIC and calculate the mean squared error for each linear model using a five time five-fold cross- validation. Each selection step (with the exception of the hdi pre-selection of parameters) now relies on this strategy.

*Finally, the combination of models from step (*5*) is nowhere detailed; are they only inspected for congruence of coefficients or are they summed to make predictions for the additive effects of two environmental stimuli?*

The combination of models, as described in Figure 1 and point 3 in subheading “Model selection”, consists of having a different linear equation for distinct partitions of the data (for example one equation for each season). They are not summed but joined together to form a piecewise linear function. To be able to compare all these composite models, we assemble all residuals calculated from cross-validation into a single vector to compute the mean squared error over the whole dataset and use the sum of the number of parameters in each equation as the total number of parameters to model the whole dataset. The MSE and number of parameters are then used to calculate the BIC of each piecewise model. These BICs are used to check whether the composite models perform better than a unique equation for the whole dataset. If so, we choose between the piecewise models the one with the lowest BIC. This is now more explicit in the Materials and methods section.

In the enrichment analysis, why were the annotations found across the three annotation resources not combined in a weighted fashion, so that GO terms more often encountered are given a higher more weight? The simple combination employed may distort the findings, particularly for the under-representation of classes (if this was tested at all).

The three resources were combined in order to increase coverage of the rice genome, not to assess the likely correctness of a given annotation. The set of rice genes covered by resource A is not the same as resource B, but neither is it disjoint. We want to achieve maximum coverage of rice genes, and thus combined the resources, but do not believe that the occurrence of the same GO annotation in multiple resources indicates a greater likelihood of that annotation being correct. Rather we consider it highly likely that the vast majority of annotations in all three resources are correct, and that double-occurrence merely indicates that a particular gene was covered by two resources.

Reviewer #2:

Finally, it would have been interesting to see how the models performed in the case of the data from Nagano et al.; rather than building models on the independent data (any strategy could have been used to this end, not only the heuristic suggested by the authors), the transferability of the models should have been assessed and commented on. This is in fact the most challenging part and would shed light on bridging the greenhouse-field gap.

Reviewer #3:

3) In addition to the technical concerns above, as written there is not enough of a comparison between this modeling approach and the one of Nagano et al. 2012 to show what this new approach has to add with respect to predicting gene expression patterns in the field. I think that the paper could benefit greatly from showing how the model selected from simple to complex would compare to using a full model including interactions with all the terms and gradually dropping terms out to compare models like in the Nagano et al. 2012 paper (although Nagano did not deal with interactions). For example, model selection on: lm(gene expression ~ genotype*season*timepoint*treatment). Building on this, I also think there is a missed opportunity here to do some mixed effects modeling (see lme4 R package) that could take into account the correlation of gene expression within and between all the treatment, genotype time of year combinations.

Due to the similarity of these two comments we address them in aggregate here.

In comparing to the Nagano et al. dataset we were attempting to do something that very few genomics studies attempt: to a compare to previous genomics studies that use different technologies. We find that nearly every genomics paper out there simply uses the latest-greatest technology and makes little or no effort to combine datasets, replicate past results or comment on the reproducibility of even the main findings. Why is this: because of the extreme difficulty of comparing the results from even slightly different technologies and different experimental designs. In this section of the paper we are trying to do this in spite of the difficulty of comparing apples and oranges. So, part of the motivation for including this comparison is our belief that the lack of comparisons to past studies is a critical void in the genomics community. All this said: we are still comparing apples and oranges (different experimental design, different technologies, different genotypes, different fields) with all the associated caveats and difficulties.

We have completely revised the way we use the data from the Nagano et al. paper to address this comment. Our new aim was to test the transferability of the results of our experiment to an independent dataset and then to compare our approach with the one developed by Nagano et al. in regard to the identification of environmental effects on gene expression and demonstrate a possible “added value”, even though the differences between the methods make this task difficult.

For the transferability aspect, our first approach was rendered awkward by the fact that we were comparing two different clustering distributions as well as comparing models that included rainfed field data and often contained several equations with models generated from only irrigated field data and comprising a unique equation. These issues made model-to-model comparison cumbersome and unclear. Therefore, we have now preferred using a new analysis of our data that includes only irrigated field gene expression and chose to only keep clusters with a single equation for both seasons of our experiment, which can be considered as a first transferability test (from one season to the other). We then used these models determined from our data to test which of the developmental/environmental effects could also be detected on a subset of the Nagano data comparable to our own data in terms of day time and plant age, organized in the same clusters as in our data. We believe that this new analysis gives interesting insights on the effect of the climatic context (tropical or temperate) on the reproducibility of environmental effects on gene expression.

We understand that it is important in assessing the impact of our work to compare it with the most similar study already conducted so far. Even though Nagano et al. use a computational approach to analyze gene expression patterns in field grown rice like we do, their experimental design and aims differ from ours so distinctively, that it makes it very difficult to compare their results with ours. First, our purpose is focused on identifying relationships between gene expression and environmental conditions while Nagano et al. are investigating all possible drivers of gene expression in order to be able to predict gene expression. Because of these distinct aims, their sampling spans all of daytime and nighttime and the whole developmental life of the plant when ours is limited to one time of day, and spans only one month of the vegetative stage. A consequence of these differences in experimental design is that we cannot apply our modeling approach to their entire dataset but need to reduce it to include only one developmental stage, one part of the day and remove circadian changes. Furthermore, Nagano et al. have adopted a very different strategy to model environmental effects: their environmental term can only consider one climatic factor per model and includes a threshold, the possibility of rectangular or sinusoidal gate and of a dose-independent response. One other major difference is that they infer models for individual genes while we do it for co-expressed groups of genes. All these differences make it impossible to conduct a valid systematic comparison of the Nagano et al. models with models that we have generated from a subset of their data. To be able to do any comparison, we could only consider extreme cases of each approach that would make the models more or less similar. We had to keep in mind however that the Nagano et al. were inferred on a much larger dataset so that any result from the comparison must be considered with caution.

Regarding mixed effects modeling, we agree with its potential, however we would first have to show the benefits of such an approach to justify the added complexity. That in itself is no easy task. And, in the context of our analysis, do we not minimize random effects by centering all data?

Reviewer #2:

At this stage, I also wonder why the profiles from different genes involved in particular process were not used to derive the models (this would have avoided the enrichment analysis as well as the clustering and might have made it for a more concrete story).

Although this is an interesting suggestion, it makes the implicit assumption that all the genes from a specific process are co-regulated, which our results do not support. This would also considerably limit the analysis compared to our whole genome approach as only a small fraction of rice genes have been assigned with certainty to specific processes.

Reviewer #3:

1) This is a large dataset with many axes of variation with one of the main conclusions being "additive and interactive effects of distinct environmental conditions on gene expression are widespread". While simplicity is nice for interpretation ("In our effort to produce simple models, we limited the number of parameters per linear equation to two"), finding the best two parameters is an arbitrary cutoff. This needs to be justified statistically by either simulations or references.

We agree that our original analysis might have been too stringent in its attempt to avoid over-fitting. We have thus run our model selection (already modified for more consistent selection on BIC only, vide supra) on the new two-season dataset with 53 clusters with no constraint on the number of parameters per linear equation. We found that among the 133 model equations selected for these 53 clusters (one cluster can be modeled with several equations for different field/season partitions), only ten contained four parameters and one had 5 five parameters, showing that equations with three or less parameters are generally enough to model the cluster means. We compared the results for the 11 clusters whose models included equations with more than three parameters with a run of model selection that constrained each equation to no more than three parameters and found that the models constrained for the number of parameters had BICs comparable to the ones with no constraint in five cases and resulted in lower BIC in two cases. The reason for this occasional inability of our method to prevent over-fitting the models resides in its using successive steps of model selection, instead of comparing all possible models at once, which is intractable with the piece-wise modeling approach we are using. In three cases, allowing for more parameters per equation did improve the model significantly, but we estimated that it was preferable to control for the over-fitting occurring in the other cases and therefore chose to limit the number of parameters per equation to three in the final version of the model selection pipeline. To put it simply, we choose to allow a few false negatives if it will prevent false positives (under the assumption that false positives trigger costly and ill-advised follow-up experiments).

Furthermore, there is discussion of interactions when these simple models do not account for interactions between variables. It is unclear from the analysis and text how interactions are defined and thus interpreted.

We agree that the use of the term “interaction” in the context of linear models can be confusing. We have therefore removed any mention of interactions in the interpretation and discussion of our modeling results to make our meaning clearer.

[Editors' note: further revisions were requested prior to acceptance, as described below.]

*The reviewers felt that the manuscript nicely shows that even though climate and local field environment have large effects on gene expression, their effects can be captured with relatively simple models. While there are not yet sufficient data to be predictive for climates or field environments that have not been evaluated yet, the finding suggests that a relatively limited amount of such data sets will enable predictive modeling. This should be emphasized in the Abstract, Introduction and Discussion.*

We have revised the Abstract, Introduction and Discussion to highlight how our study can contribute to the undertaking of predicting transcriptional patterns.

Reviewer #2:

*The authors have carefully considered and addressed the comments raised by the reviewers with respect to the selection of number of clusters (already a non-trivial task, given the number of different measures they considered), number of piecewise models (per BIC only), and averaging of profiles. However, the issue with the model simplification by pruning correlated parameters seems to differ between the response and the main text. More specifically, it is not clear how the decision in paragraph two, subheading “Modeling the effect of climatic factors on transcriptomic variation in different field environments” was implemented, as this can be carried out in many ways. This statement does not correspond to the response provided in your letter, whereby the correlation is set to 0.98 and the parameters were averaged (this strategy of averaging provides a deterministic and reproducible outcome).*

This concern arises from a confusion between two different issues. First, the pruning of the correlated parameters was done as described in the first response letter and described in the Materials and methods section. Second, after increasing the number of parameters that could be included in each equation, we had to deal with the problem that some equations comprised parameters that were correlated with each other (though not with r > 0.98, as these were already averaged together), which increased the risk of over-fitting without being very informative. This was remedied by allowing only one of a pair of correlated parameters to appear in an equation. The choice was also deterministic, as the chosen parameter was the one that resulted in the best model.

*In paragraph four, subheading “Modeling the effect of climatic factors on transcriptomic variation in different field environments”, it should be included that the BIC is calculated on the joint piecewise models; I suppose the number of parameters corresponds to the number of predictors summed over all models, or does it correspond to the number of unique predictors over all models? This will have effect on the final findings.*

We have made the suggested modification. The BIC is calculated using the sum of number of parameters over all models, because if the same parameter appeared in several equations, it would be with a different coefficient, contributing to increasing model complexity as penalized by the BIC. This has been made clearer in the Materials and methods section.

*I would suggest that the claim about "designing model selection approach" is toned done, as the authors themselves state in the response letter that "we do not claim to provide any notable advances in this paper for the computational community". While this may be an article aimed for the genomics-enabled biologist, it will be read by computational biologists interested in the large data resource provided.*

We have rephrased this sentence to “We used a model selection approach (…)”.

*Finally, while the authors provided a pipeline based on well-established methods, the interested reader may want to know what is the added value with respect to now classical approaches (e.g., mixed effect models), as one of the other reviewers already suggested.*

The major advantage of our method is its ability to examine a large number of parameters and select only the sparse models. This allows an extensive examination of environmental and developmental parameters. More traditional variance methods (such as mixed model methods) are difficult to implement in this context because of the need for these approaches to put all of the parameters simultaneously in the models, which results in over-fitting.

*Rephrasing of some key sentences may be needed, to fully match what was stated in the main text and response letter: 1) The statement in the Abstract "we show that new insights can be gained" should indicate what these insights precisely are.*

We have reworked the Abstract to more specifically refer to multiple biological insights resulting from our study.

*2) The impact statement should be clarified – one can determine a model for any set of variables; however, the pressing problem is to show that the model explains a major part of the variance and can be used for predictive purposed. Therefore, it also needs rephrasing.*

The impact statement has been changed accordingly.

Reviewer #3:

*1) In re-reviewing this paper the authors have done a great deal to clarify the modeling approach and have provided many details that were requested. I think that they have made great improvements in the technical aspects of this paper. However, now that the modeling details are clearer the manuscript could be vastly improved for readability for a biological audience (as a stated goal in response to reviewer 1's comment) if the biological question was clearly outlined in the Introduction and followed through the Methods, Results and Discussion. In the Abstract: "We show that new insight can be gained from studying the effects of co-occurring abiotic stimuli in complex dynamic environments." Now that the technical aspects of the methods are clearer I am struggling to see what new biological insight is gained from this paper. My main criticism of this paper as a whole in this new version is that it reads as a new way to do exploratory data analysis in a large-scale field transcriptomics experiment, but is not framed to ask a direct biological question with the data/analysis.*

We have extensively revised the Abstract, Introduction and Discussion, and some of the writing in the Results to frame our approach around two biological questions:

1) How are multiple fluctuating environmental signals integrated in transcriptional responses in the field?

2) How does context (e.g., season, field type) impact the climatic responses of gene expression?

For each of these questions a paragraph of the Introduction describes the issues and current knowledge, and a section of the Discussion gives the answers that our results provide. We have also highlighted in the Abstract additional insights not directly related to these questions. Finally, in our Conclusion section, we describe the advance our study provides in the efforts to develop models of gene expression, and the implications of such work on understanding species response to climate change and on crop design.

*In my last review I stated: "as written there is not enough of a comparison between this modeling approach and the one of Nagano et al. 2012 to show what this new approach has to add with respect to predicting gene expression patterns in the field." I followed this with some suggestions as ways to approach this using a mixed model approach that could be applied to genes or gene clusters of interest. The authors have done extra work to apply their method to the Nagano et al. 2012 paper, but I still do not see what new biological insight is gained from the cluster method over gene-wise predictions. If the truly novel aspect of this dataset (compared to Nagano et al. 2012) is the drought/non-drought comparison then could the paper focus much more on this comparison?*

The rationale for the clustering step is more methodological than conceptual. Our results show that reproducibility is much higher for cluster means than individual genes. This means that the environmental expression response of most genes in our analysis is confounded by stochastic effects and responses to endogenous signals, that we cannot model. Fitting models to these expression profiles would therefore greatly increase the risk of fitting equations that have no causal relationship with transcript accumulation. We found it more informative to cluster all genes and work on reproducible cluster means than only model a few genes with reproducible patterns of expression. As explained before, the novel aspects of this dataset compared to Nagano et al. is that we are assessing the integration of multiple environmental signals (including drought) and the effect of different types of environmental context.

*2) Interactions implied in Introduction and Conclusion, but not tested in models:*

*Introduction: “In addition to the dynamic nature of field conditions, the interaction between multiple stimuli is another major cause of discrepancies in plant responses/phenotypes between laboratory and field conditions.” Conclusion: "By working in natural, complex conditions in the field, we can examine interactions between the effects of factors that vary at distinct time-scales, like temperature and water availability." In the Authors’ response: "We agree that the use of the term "interaction" in the context of linear models can be confusing. We have therefore removed any mention of interactions in the interpretation and discussion of our modeling results to make our meaning clearer." Although the authors disagreed with me about the including models with interaction terms for the entire dataset, I still think it would be a good approach to make distinct comparisons between genes/gene clusters and how they respond to the drought treatment. For example, in the subsection “The agricultural field environment strongly affects transcriptional responses to climatic fluctuations”, the authors state: "This result suggests that, in our experiment, a group of abiotic stress response genes responded to a greater extent to high light/heat stress than drought stress, while another group of abiotic stress response was much more sensitive to the effect of drought." Why not subset the data based on this observation and test these comparisons directly with a follow up model that includes an interaction term?*

After the first reviews, we understood that for readers with a modeling perspective, “interaction” refers to a precise term in an equation, which is why we have removed it from the interpretation and discussion of all of our results. However, we thought it would be acceptable to keep the word “interactions” in the Introduction and Discussion in reference to previous work about the effect of combined stresses where this word was used. In these studies, interaction did not refer to an interaction term in a model but to the fact that the effect of co-occurring stresses cannot be inferred from single stress treatments. As there is still confusion in this regard, we have rephrased these findings to make clearer what these studies meant, without using the word interaction.

Because of the tools we use for model selection, to integrate interactions into our modeling approach, we would have to add interactive parameters for each pair of environmental and developmental parameter, which would considerably inflate our number of parameters (from about 70 to several thousand). This would certainly lead to over-fitting our expression data (given that we only have a few hundred RNA-seq observations and they are not independent; e.g. are composed in time series). This explains why we did not choose to include interactions terms in our equations. There are however implicit interactions in the models where different subsets of the data have different models: a cluster modeled with different equations in the irrigated and rainfed fields underlies the existence of an interaction between climatic effects and the field environment.

We agree that the example from the subsection “The agricultural field environment strongly affects transcriptional responses to climatic fluctuations” cited here was confusing and hinted at different types of interactions without making them explicit. This part of the Discussion has been removed, as it did not fit within the biological questions our article is now structured around. A follow-up model on this matter is therefore not relevant anymore.

*3) Do these gene expression clusters mean anything for the plant as far as other phenotypes are concerned? There is a missed opportunity here to expand what is found in the clusters into a story about how the environment influences gene expression and how that gene expression might influence plant growth/development. The Discussion starts to get at this, but the message is lost in the details about what a few genes from each cluster mean. The clustering/model selection approach is a great way to reduce this complex dataset to fewer significant parameters but as this manuscript is written does not help in biological interpretation of the clusters. This is also where a clear biological goal of the modeling/method/experimental design is necessary and could improve this manuscript.*

Even though we did measure some developmental phenotypes (number of tillers, plant height, flowering time) during our experiments, it has not been possible to relate them with our transcriptomics results. While we did not measure significant differences in transcriptional patterns between genotypes, the two landraces showed differences in morphology and phenology of the same range as the between-field differences, and averaging the genotypes like we did for expression would confound the field and season differences. With a more appropriate experimental design and more detailed phenotypic data, relating gene expression with specific phenotypes in natural conditions would be of great interest but it is outside the scope and possibilities of our study.

Nevertheless, we do agree that the functional analysis of our clusters should have been more thoroughly exploited. Therefore, we have added a section to our Discussion where we examine some biological implications of our GO enrichment results.